# Flame retardant high-power Li-S flexible batteries enabled by bio-macromolecular binder integrating conformal fractions

Chenrayan Senthil [1], Sun-Sik Kim[1] & Hyun Young Jung [1,2 ✉]

Polymer binders for sulfur cathodes play a very critical role as they prerequisites for an in-situ immobilization against polysulfide shuttle and volume change, while ensuring good adhesion within active materials for ion conduction along with robust mechanical and chemical stability. Here, we demonstrate anionic surface charge facilitated bio-polymer binder for sulfur cathodes enabling excellent performance and fire safety improvement. The aqueous-processable tragacanth gum-based binder is adjusted to house high sulfur loading over 12 mg cm$^{-2}$ without compromising the sulfur utility and reversibility, imparting high accessibility for Li-ions to sulfur particles about 80%. The intrinsic rod and sphere-like saccharidic conformal fraction's multifunctional polar units act as active channels to reach the sulfur particles. As a result, the binder entraps polysulfides with 46% improvement and restrains the volume changes within 16 % even at 4 C. Moreover, the flexible Li-S battery delivers a stack gravimetric energy density of 243 Wh kg$^{-1}$, demonstrating high reactivity of sulfur along with good shape conformality, which would open an avenue for the potential development of the compact and flexible high-power device.

---

[1] Department of Energy Engineering, Gyeongsang National University, Jinju-si, Gyeongnam 52725, South Korea. [2] Future Convergence Technology Research Institute, Gyeongsang National University, Jinju-si, Gyeongnam 52725, South Korea. ✉email: hyjung@gnu.ac.kr

The ease of access and a highly facilitated diffusion of Li-ions to sulfur particles within the cathode are crucial to increase the electrochemical reactivity of the sulfur to realize performances approaching the theoretical energy density of 2567 Wh kg$^{-1}$ (refs. [1–3]). However, critical concerns revolve around sulfur as cathode hosts are segmental; its intrinsically insulative ($5 \times 10^{-30}$ S cm$^{-1}$ at 25 °C) nature and the electrode deterioration due to the particle stress-related volume changes and the solubility of intermediate polysulfides ($S_{4–8}{}^{2-}$) during electrochemical cycling hindered the commercialization of practical Li-S batteries[4–6]. Hence, to maximize the sulfur cathode's performance and cycle life, an ideal binder requires a wide electrochemical window, good electrolyte wettability, low electrode polarization and charge-transfer resistance, strong chemisorption towards polysulfides, cycling stability, and mechanical stability.

Initially, the role of binder was limited to the fundamental electrode architecture such as interparticle contact and adhesion when linear polymers[7–9] were used, and later the eventual need for mechanical strength was fulfilled through the functional groups containing linear polymers[10–12]. However, linear polymeric materials, including commercial polyvinylidene difluoride (PVDF) and recently studied poly(ethylene oxide) (PEO), still have limitations, such as low sulfur utilization, polysulfide dissolution, poor Li-ion conduction, or low mechanical strength (Fig. 1b). Such concerns gave rise to the development of crosslinked[13,14], linear-anchor binders[15–17] or hybrid binders[18–20] that contain multiple polar functional groups designed to restrain the volume changes of sulfur (up to 80%) and immobilize polysulfides ($Li_2S_n$, $n = 4–8$) during cycling. Advertently, the role of binder has also extended in combination with sulfur cathodes such as sulfur-rich polymers[21,22], sulfur host materials, redox mediators[23,24], and embedded current collectors[25] to achieve long-term stability, however, with limited accomplishments.

Here, we elucidate a highly branched biopolymer that integrates abundant polar functional groups distinct from other binders through its conformal molecular properties. The complex and macromolecular biopolymer reported as a binder for the first time in Li-S battery, *Astragalus gummifer* gum (Tragacanth gum, TG), inherits heterogenous polysaccharide fractions which mitigate three critical prerequisites for a better operation of high-performance Li-S battery; (1) enables high access for Li-ions to sulfur-active particles (up to 80.3% of S) through the anionic polymeric backbone moieties triggering efficient sulfur reactivity, (2) exposed polar functional groups of saccharidic units expedite trapping of soluble polysulfides ($Li_2S_n$, $n = 4–8$) into the electrolyte to maintain its concentration, and (3) actively regulate large volume changes through stretchable rod-like tragacanthin and sphere-like bassorin molecular conformations ensuring robust mechanical strength (Fig. 1a and Supplementary Figs. 1, 2). Apart, the water-processable and fire-retardant ability of TG binder impart environmentally compliant and good thermal stability for safer Li-S batteries. The prospective findings from our study underpin multifunctional positive attributes of the TG binder in realizing high-performance, flexible and safer cathodes for Li-S batteries. Therefore, these definite improvements in the sulfur battery chemistries would pave the way to realize high-performance electrochemical energy devices for e-transportation, portable power electronics, and so on, considering the prevalence, cost-effective and sustainable characteristics of sulfur[7–11,17–22].

## Results

**Reactivity of sulfur and lithium in the binders.** The TG-driven enhanced electrochemical kinetics of sulfur cathodes, i.e., reversible interconversion of $S_8/Li_2S$ was confirmed through cyclic voltammetry (CV), galvanostatic charge-discharge (GCD), and electrochemical impedance spectroscopy (EIS) measurements performed using the Li-S cells that employ either PVDF, PEO, or TG as the binder for sulfur cathode (Fig. 1c–e). The redox kinetics accessed through CV studies expose typical reduction and oxidation characteristics representative of elemental sulfur reduction ($S_8$) to high-order polysulfides ($Li_2S_8 \rightarrow Li_2S_6 \rightarrow Li_2S_4$) and then to low-order polysulfides ($Li_2S_4 \rightarrow Li_2S_2 \rightarrow Li_2S$) noticeable as cathodic peaks between 2.4–2.2 V (C1) and 1.8–1.9 V (C2), respectively (Fig. 1c). Following, the anodic scan reverses as formed lithium sulfides ($Li_2S_2/Li_2S$) to lithium polysulfides (LiPS)/sulfur notable as an overlapped peak about 2.4–2.6 V (A1) (refs. [10,17,26]). Although these processes fairly occur, efficient conversion of $S_8$ to $Li_2S$ and LiPS/sulfur notable as high peak currents and smaller overpotentials indicates the superiority of the S/TG, where the current was eventually twice to four-fold times more than S/PVDF and S/PEO electrodes. Thus, the notable increase in the kinetics of S/TG is exclusively accounted for higher sulfur utilization during the redox process, along with facile interconversion of $S_8$ to $Li_2S$ species. Meanwhile, it is worth mentioning that pure TG as electrodes does not contribute to the capacity (Supplementary Fig. 3); thus, the electrochemical participation of the TG itself is ruled out.

Further, insights into the electrochemical utilization of sulfur and the improved cyclability were revealed through GCD experiments (Fig. 1d). At a current rate of 1 C (1 C = 1675 mA g$^{-1}{}_{sulfur}$), the discharge capacity observed at the 100$^{th}$ cycle was much higher for the S/TG electrode than the S/PVDF and S/PEO electrodes, where the capacity of the sulfur electrode using TG binder stood as high as 391 and 142 mAh g$^{-1}$ than the conventional PVDF and PEO binders. These significant differences in the reversible capacities denote that the electrochemical sulfur reduction is more efficient to about 37 and 28% in the TG binder when compared with the PVDF and PEO binders. The prospect of voltage polarization ($\Delta V$) upon efficient sulfur utilization during the discharge-charge cycle was found to be smaller at the S/TG electrode with $\Delta V_{TG} = 0.18$ V against $\Delta V_{PVDF} = 0.28$ V for S/PVDF and $\Delta V_{PEO} = 0.43$ V for S/PEO electrodes. Such a minimal $\Delta V$ implies lower resistance of TG binder to cause favorable access and diffusion of Li-ions to react with the sulfur particles even beneath the carbon and also offers an efficient ion transport across the sulfur/TG interface. Moreover, the experimental electrode polarization can be deliberated to understand the sulfur reaction kinetics reflected through a series of plateaus during the discharge-charge process. The characteristic upper and lower voltage regions observed as plateaus and slopes during the discharge of S/TG electrodes extended far more to a significant capacity than those from S/PVDF and S/PEO electrodes, indicating the lower resistance and barrier in TG for promoting a series of chemical transformations; solid-state sulfur ($S_8$) to liquid-state sulfur ($S_4{}^{2-}$) and insoluble sulfides ($Li_2S$) giving rise to an additional capacity. The lower resistance and polarization were further confirmed through the Nyquist plots obtained from EIS measurements in Fig. 1e. Characteristic Nyquist plots show an intercept at a high-frequency region ($R_s$, ohmic resistance), a semicircle at medium-frequency ($R_{ct}$, charge-transfer resistance), and an inclined line at low-frequency region ($Z_s$, Warburg impedance) to the real axis. Typically, all the electrodes exposed similar solution resistance ($R_s$ for S/PVDF = 10.06 Ω, S/PEO = 6.4 Ω, and S/TG = 7.9 Ω); however, the $R_{ct}$ for the S/TG electrode was smaller as 125.19 Ω, whereas higher values were evident for S/PVDF at 249.34 Ω and S/PEO at 211.69 Ω, indicating feasible binder interface for active sulfur and Li-ion interaction to offer improved ionic and electronic conductivity in S/TG electrodes.

In Fig. 2, the significant electrochemical performance gains of Li-S batteries employing sulfur cathodes fabricated using TG binders were evaluated in detail by comparing PVDF and PEO

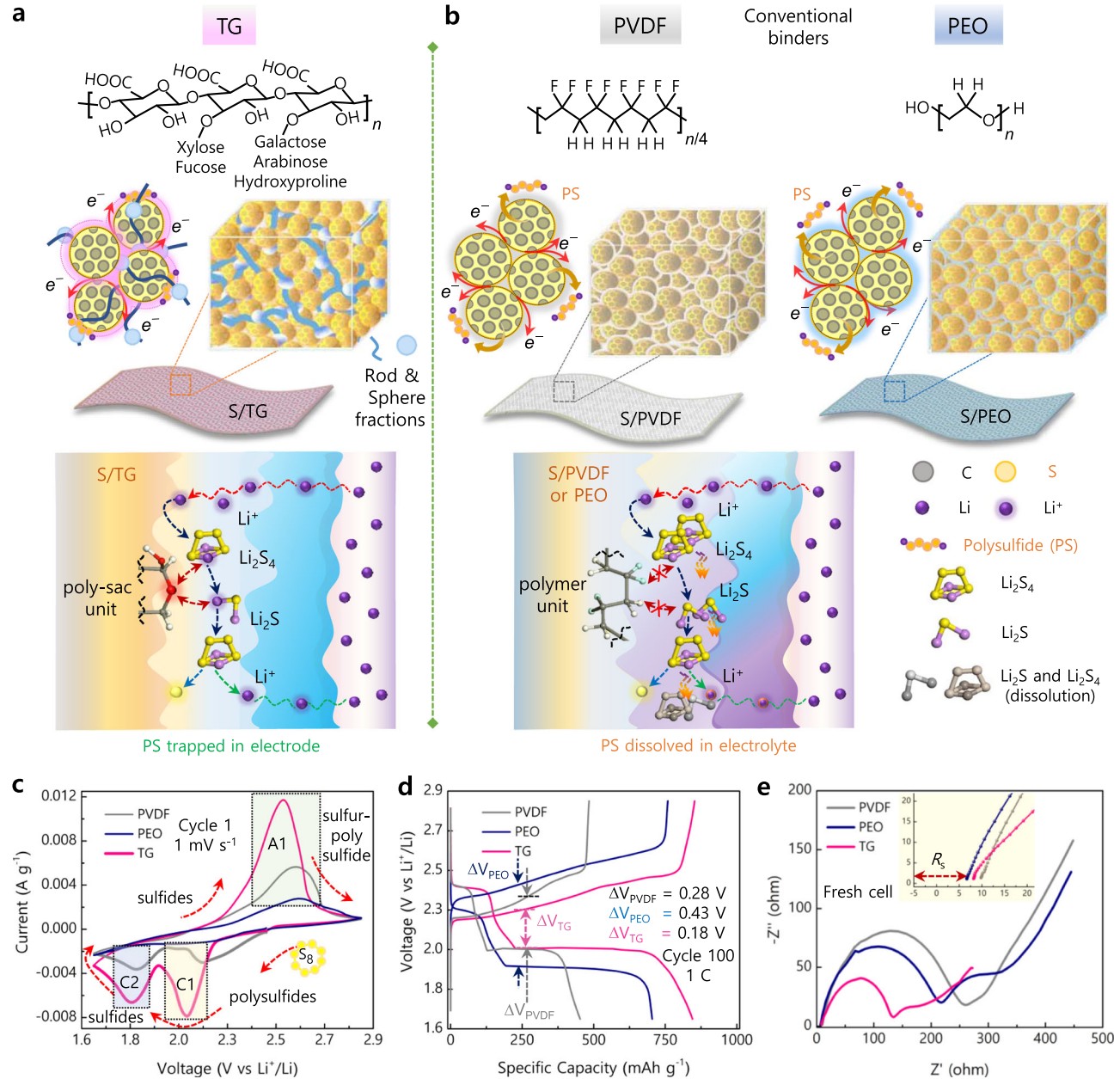

**Fig. 1 Schematic illustration of the fabrication, architecture, and properties of various binders for sulfur cathodes. a, b** The cathode is comprised of sulfur loaded into 'Super P' carbon with binders. **a** Tragacanth gum (TG). **b** Polyvinylidene difluoride (PVDF) and poly(ethylene oxide) (PEO). **c–e** Electrochemical performance of sulfur cathodes fabricated using TG as the binder, S/TG electrodes against the PVDF, S/PVDF, and PEO, S/PEO for Li-S battery. **c** Cyclic voltammograms recorded between 1.7 and 2.8 V (Li$^+$/Li) at a scan rate of 1 mV s$^{-1}$. **d** Galvanostatic discharge-charge profiles measured at a current rate of 1 C (1 C = 1675 mA g$^{-1}$). **e** Nyquist plots for the fresh Li-S cells employing various binders.

binders. The CV studies were performed in the voltage window 1.7–2.8 V (vs. Li$^+$/Li) at a scan rate of 1 mV s$^{-1}$ (Fig. 2a–c). The cathodic peaks (C1 and C2) were polarized at 2.34 V and 1.94 V for S/TG against 2.34 V and 1.91 V for S/PVDF and 2.25 V and 1.88 V for S/PEO electrodes, respectively. A similar fingerprint was represented in the anodic scan at 2.48, 2.53, and 2.54 V for S/TG, S/PVDF, and S/PEO, implying the improved kinetics and conductivity offered by the TG binder. This characteristic is further manifested in the peak shift ($\Delta V_x$) observed upon successive reversibility of the electrodes. It was found that the cathodic ($\Delta V_1$ and $\Delta V_2$) and anodic ($\Delta V_3$) peak shifts were, respectively, 0.019, 0.029, and 0.021 mV for the S/TG electrode, which was a smaller difference compared to 0.061, 0.053, and 0.067 for S/PVDF and 0.042, 0.053, and 0.087 mV for S/PEO

electrodes. These differences demonstrate the significant impact of TG binders to promote reversible interconversion and efficient reduction of sulfur during the Li-S battery cycling. Apart from the initial cycle, where the reduction peaks are shifted due to dominant activation processes[27] as owed to high sulfur reactivity, the subsequent scans show overlapped cathodic (C1 and C2) and anodic (A1) peaks indicative of excellent cycling stability in the S/TG electrode. On the contrary, the S/PVDF and S/PEO electrodes still showed a gradual diminish and shift in the redox profile as the scans proceeded due to their decreased access to sulfur particles[8,20].

Galvanostatic discharge-charge profiles were obtained at a current rate of 1 C, and an in-depth analysis reveals several fingerprints desirable to understand the electrode kinetics

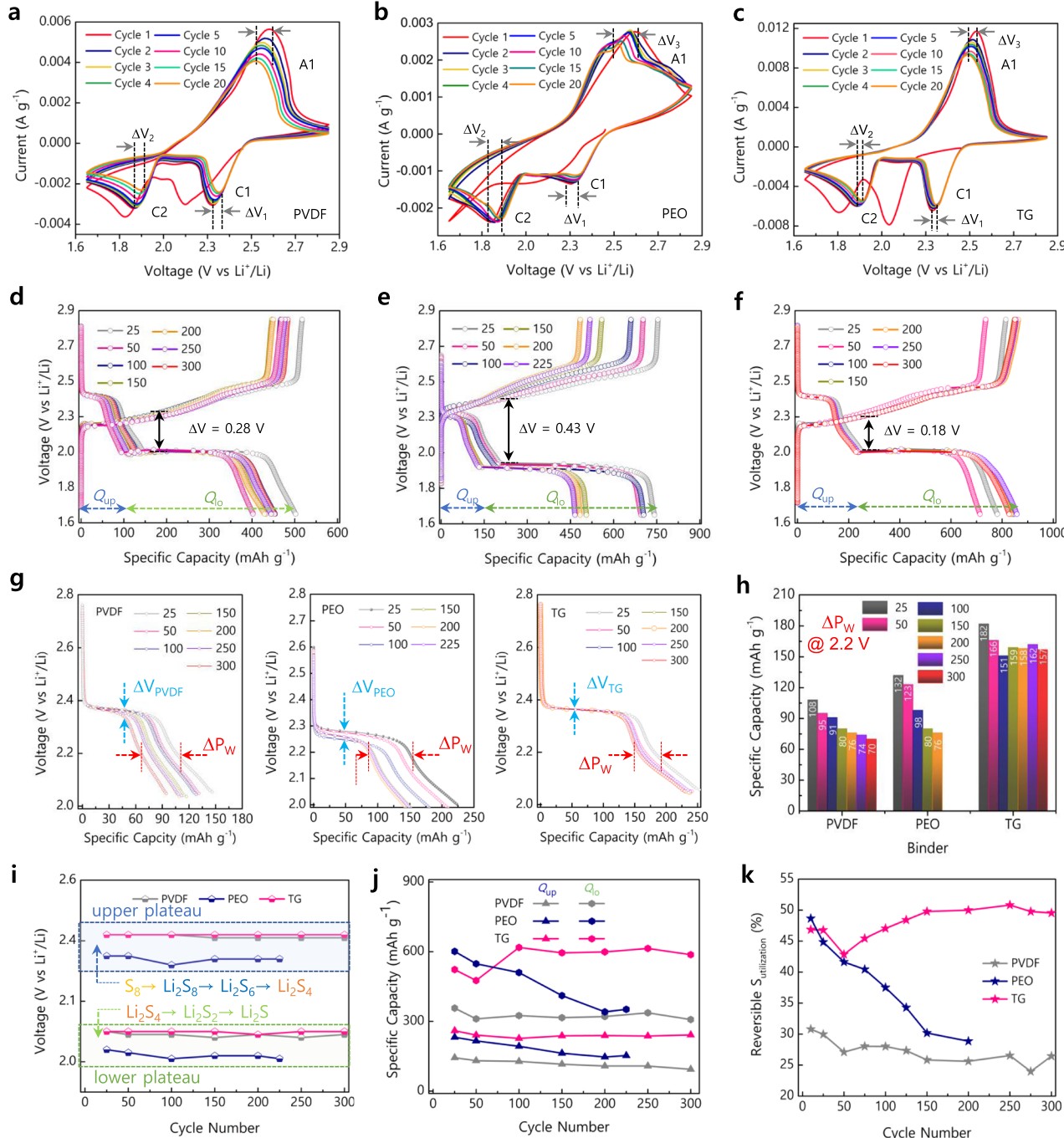

**Fig. 2 Improved electrokinetic properties of S/TG compared with S/PVDF and S/PEO cathodes. a–c** Cyclic voltammetry (CV) curves recorded at a scan rate of $1\,mV\,s^{-1}$. **d–f** Discharge-charge profiles at a current rate of $1\,C$ ($1.675\,Ah\,g^{-1}$). **g** Onset potential at different cycles extracted from **d–f**. **h** Corresponding capacity observed at 2.2 V. **i** Onset potential denoting the reduction of sulfur in the upper and lower plateau regions during cycling at $1\,C$. **j** Capacity fade and retention at the respective upper potential ($Q_{up}$) and lower potential ($Q_{lo}$) extracted from **d–f**. **k** Reversible sulfur utilization of various electrodes shown for selected cycle number.

(Fig. 2d–g). From the typical voltage profile, the discharge curve represents a multistep sulfur reduction to $Li_2S$ upon accepting Li-ions and electrons as apparent through the plateaus at high and low voltage regions[6,7]. In Fig. 2d–f, the range to which the discharge profile extends and remains stable exposes the higher deliverable capacity and superiority of S/TG among S/PVDF and S/PEO electrodes. In such a case, the S/TG electrodes exhibited a high discharge/charge capacity of $829/856\,mAh\,g^{-1}$ at the end of the 300th cycle, which is attributed to an efficient sulfur reduction and utilization. It is worth mentioning that the highly

reversible and stable capacity of the S/TG electrode is exclusively achievable only in conjunction with the lower resistance of the electrode constituents (Figs. 1e, 2f). To substantiate this claim, the voltage polarization ($\Delta V$) taken as a measure of resistance exerted during the discharge-charge cycle was found to be higher for S/PEO and S/PVDF electrodes as 0.43 and 0.28 V, respectively (Fig. 2d, e). In comparison, S/TG showed a lower polarization of 0.18 V due to nominal resistance that provides good stability and integrity of electrode particles by the TG binder (Fig. 2f).

Further, insights into the polarization and resistive properties are depicted in Fig. 2g, h expressed as $\Delta V_x$ ($x$ represents either PVDF, PEO, or TG), where one can visualize the outstanding voltage stability of the TG binder. As the cycles progressed, the $\Delta V_x$ was well-overlapped for S/TG than the S/PVDF and S/PEO electrodes (Fig. 2g). Apart, the discharge voltage regimen distinguished as the upper and lower plateau was maintained stable for the S/TG electrodes at potentials 2.36 and 2.05 V (Fig. 2i) even with an increase in the cycles. Comparatively, the S/PEO electrodes exhibited a wide variation in the upper plateau regions suggesting the decreased kinetics of active sulfur owed to more polysulfides generation. It is worth mentioning that the exceptional voltage stability of the TG binder stems from the preservation of electrode environment such as electrode integrity, electrolyte concentration, controlling irreversible losses during cycling. In addition, the profile width ($\Delta P_w$), a phenomenon exclusively reported in this study, was measured at a discharge voltage of 2.2 V to determine the irreversible capacity losses as a function of voltage, which remained relatively much shorter and concise for S/TG electrodes. This parameter $\Delta P_w$ is necessary to elucidate the shuttle effect that occurs during the charging process as a consequence of previous discharge. The observed capacity losses for selective discharges ranging from 50 to 200 cycles were within a variable of 8 mAh g$^{-1}$ for S/TG, comparing S/PVDF of 22 mAh g$^{-1}$ and S/PEO electrodes of 47 mAh g$^{-1}$ (Fig. 2g, h). Such a minimal $\Delta P_w$ variable for TG is only possible upon efficient trapping and reversibility of polysulfides as the cycle progresses without paving the way for the detrimental shuttle effect that is well known to induce capacity decay as prevalent for either S/PVDF or S/PEO electrodes.

Understandings from the electrochemical cycling profile, the conversion of sulfur particles (solid) to soluble sulfur (liquid) and its reduction to $S_4^{2-}$ observed as plateau at higher voltage and a slope extending till 2.0 V (since the value varies with the binder, knee point is considered) contributes to about half-electron, i.e., one-fourth of the total electron ($0.5e^-$ out of $2e^-$) transfer per sulfur atom accounting 25% of the sulfur's theoretical capacity (1675 mAh g$^{-1}$ vs. Li$^+$/Li)5[28]. Noticeably, in Fig. 2j, the respective process denoted as $Q_{up}$ (capacity of the upper potential, >2.0 V) calculated at the end of 200th discharge contributed to about 108, 142, and 239 mAh g$^{-1}$ for the sulfur electrodes using PVDF, PEO, and TG binders, respectively, which is 14.26% for TG as compared to the theoretical value (25%, 419 mAh g$^{-1}$ against 1675 mAh g$^{-1}$), while S/PVDF and S/PEO electrodes showed 6.44% and 8.47% sulfur utilization. Besides, upon the depth of discharge (DOD) to 1.7 V, the soluble liquid-state $S_4^{2-}$ is further reduced to insoluble Li$_2$S$_2$/Li$_2$S observed as a lower plateau and slope contributes the rest 75% capacity as documented previously; this process is denoted as $Q_{lo}$ (capacity of the lower plateau, <2.0 V) revealed the experimental lithium sulfide conversion was higher as 35.94% for S/TG electrode against 19.16% for S/PVDF and 20.48% for S/PEO. Upon considering the sum of capacity contributions ($Q_{up} + Q_{lo}$), the efficient interconversion of polysulfides and reduction of sulfur was found to be higher as 50.2% for S/TG against 25.6% of S/PVDF and 28.95% of S/PEO electrodes at the end of the 200th cycle (Fig. 2k). This indicates that in S/PVDF or S/PEO electrodes, about three-fourth to two-third of sulfur is lost or remained as dead sulfur, which clearly manifests strong evidence for the electrochemically improved sulfur utilization and polysulfide conversion in S/TG electrodes.

The long-term cyclability of sulfur electrodes was validated through galvanostatic cycling performed at the current rate of C/5 (Fig. 3a). Noticeably, the higher reduction of sulfur for the S/TG electrode is evident even from the initial discharge process delivering a capacity of 1239 mAh g$^{-1}$, whereas 813 mAh g$^{-1}$ and

1109 mAh g$^{-1}$ was obtained for S/PVDF and S/PEO electrodes, respectively, denoting the sulfur reduction/utilization is about 73.9% for S/TG against 48.5% and 66.2% for S/PVDF and S/PEO electrodes. Further, as the cycles progressed, both the S/PVDF and S/PEO electrodes experienced the shuttle effect of losing active sulfur particles, as discussed later, thereby leading to a decreased reversible capacity of 392 and 539 mAh g$^{-1}$ at the end of the 100th cycle. However, the S/TG electrodes witnessed a stable and higher reversible capacity of 976 mAh g$^{-1}$, representing successful polysulfide trapping and reversible capability. It should be noted that the S/TG electrodes reversibly utilized 78.77% (against the first cycle, 73.9%) of sulfur even at the end of the 100th cycle, whereas relatively 48.21% (48.5%) and 53.41% (66.2%) were accessible for S/PVDF and S/PEO electrodes, respectively (Fig. 3a, h).

Furthermore, the cycling stability at higher current rates of 1 C and 4 C was investigated (Fig. 3b, c). At a current rate of 1 C (Fig. 3b), the S/TG electrodes witnessed a steady and stable discharge capacity >821 mAh g$^{-1}$ after the 8th cycle, and even after 225th cycles, the discharge/charge capacity stood at 832/865 mAh g$^{-1}$. For the S/PEO and S/PVDF electrodes, the capacity retention was lowered to 58 and 80% against an incremental capacity of 104% for S/TG electrodes. Moreover, the capacity fade for S/PEO and S/PVDF electrodes results from a marginal increase of insoluble polysulfide accumulation as the number of cycles increases. Such depositions are considered dead sulfur which causes an increase in the electrode resistance, thereby limiting the transport of Li-ions and electrons. The high-rate capability of the electrodes was evaluated by GCD experiments performed at a current rate of 4 C (Fig. 3c) and rate capability studies (Supplementary Fig. 4). It is widely documented in battery chemistries that the electrodes subjected to high current rates increase the cell resistance as the diffusion and movement of ions are hindered by the interparticle resistance[29]. Nevertheless, the S/TG electrodes showed a higher discharge capacity than the S/PVDF and S/PEO electrodes, along with a stabilized capacity notable after 23 cycles. The practical capacity of S/TG electrodes attained 406 mAh g$^{-1}$ at the end of the 1000th cycle, whereas the S/PEO and S/PVDF electrodes showed 127 and 43 mAh g$^{-1}$, respectively. It is worth mentioning that the TG binder can offer plentiful access to sulfur particles and provide space for ionic and electronic transport even at high current rates. The sulfur utilization for S/TG electrodes at various current rates stood at 78.3%, 69.4%, and 54.0% for C/5, 1 C, and 4 C, respectively (Fig. 3i). Moreover, the shuttle factor ($f$) calculated for the S/TG electrodes at the end of the 1000th cycle at 4 C was 0.07 in $C_{eff}$ of 98.1%, and relatively $C_{eff}$ of 93.8% and 92.1% for S/PVDF and S/PEO electrodes (Supplementary Fig. 5) led to $f$ values of 0.13 and 0.19, respectively (see a section for shuttle factor calculation in the supplementary). Here, $f$ approaching zero indicates a lower shuttle effect; in such case, the lower $f = 0.07$ clearly manifests the suppression of polysulfide shuttle, which eventually leads to a stable discharge/charge capacities of S/TG electrodes than the S/PVDF and S/PEO. To emphasize, the low shuttle factor of the S/TG is concomitant with the relatively minimal $\Delta P_w$, high capacity generated at $Q_{up}$, and high sulfur utilization, as discussed earlier. Thus, the TG binder appears to have effectively coped with critical concerns of Li-S batteries to make the S/TG electrodes outperform other electrodes, particularly efficient sulfur utilization, polysulfide trapping, and mitigating the volume changes.

The practical prospectus of the TG binder in terms of commercial Li-S cells was evaluated by fabricating highly loaded sulfur electrodes. Fabricating high sulfur content TG electrodes, especially mass loadings above 8 mg cm$^{-2}$, witnessed minor cracks over the electrode surface (Supplementary Fig. 6). In order to overcome this issue, we propose "gelation", a swellable-gel-like

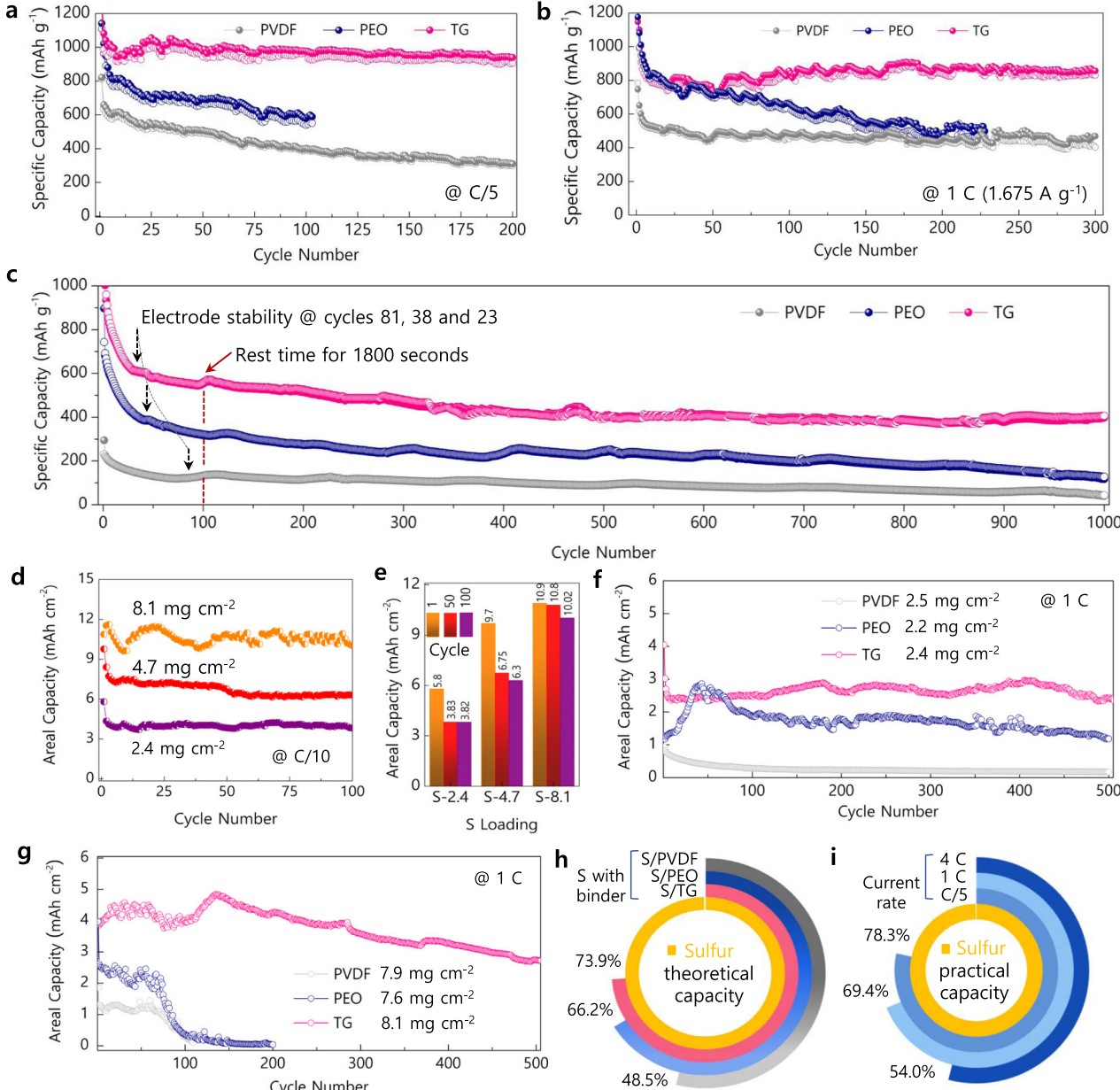

**Fig. 3 Capacity retention and sulfur utilization analysis of sulfur cathodes with PVDF, PEO, and TG binder. a–c** Cycling performances obtained at current rates C/5, 1 C, and 4 C, respectively. **d, e** Cycling performance with various sulfur loading using TG binder. **f, g** Long-term cycling stability for high sulfur-loaded S/PVDF, S/PEO, and S/TG electrodes at a current rate of 1 C. **h** Sulfur utilization for various binders at C/5 (against a theoretical capacity of S = 1675 mAh g⁻¹). **i** Sulfur utilization for S/TG electrodes at various current rates of 4 C, 1 C, and C/5 (against a practical capacity of S).

viscous slurry of TG binder was initially prepared by prolonging the duration of slurry formation and used to disperse the active sulfur/carbon mixture through gentle addition (90–120 min) and then laminated. The resultant method allowed the loading mass of sulfur to successfully exceed even above 12 mg cm⁻² with a tap density of S/TG powders at 1.08 g cm⁻³. The S/TG electrodes of 2.4, 4.7, and 8.1 mg cm⁻² were evaluated at a current rate of 0.1 C (Fig. 3d), where the electrodes of mass 2.4 and 4.7 mg cm⁻² exhibited high areal capacities of 5.8 and 9.7 mAh cm⁻² and remained steady at the end of 100 cycles, which is comparable and higher than the reported capacities achieved using aqueous-based binders (Supplementary Table 1). The capacity decay calculated between the 3rd and 100th cycle was 0.007% and 0.009%, respectively (Fig. 3e). Furthermore, the TG electrodes of 8.1 mg cm⁻² delivered a high areal capacity of 10.02 mAh cm⁻²

with the capacity fade of 0.015% that is slightly higher than the 2.4 and 4.7 mg cm⁻² electrodes (Fig. 3e). Besides, the robust nature of high sulfur-loaded TG electrodes is prominent from the long-term cycling stability shown in Fig. 3f, g. The S/TG cathodes tested with sulfur loading 2.4 mg cm⁻² exhibited an areal capacity of 2.4 mAh cm⁻² and demonstrated excellent cycling stability featuring 95.2% capacity retention even after 500 cycles. Under similar conditions, the S/PVDF and S/PEO electrodes of mass 2.5 and 2.2 mg cm⁻² exhibited discharge capacities of 0.17 and 1.18 mAh cm⁻², respectively, with severe capacity fade. The case is more for high sulfur-loaded S/PVDF and S/PEO electrodes, while S/TG showed a marginal decay. This observation could be applied to develop robust and high active mass loading sulfur cathodes as Li-ion diffusion and transport are influenced by the electrode thickness.

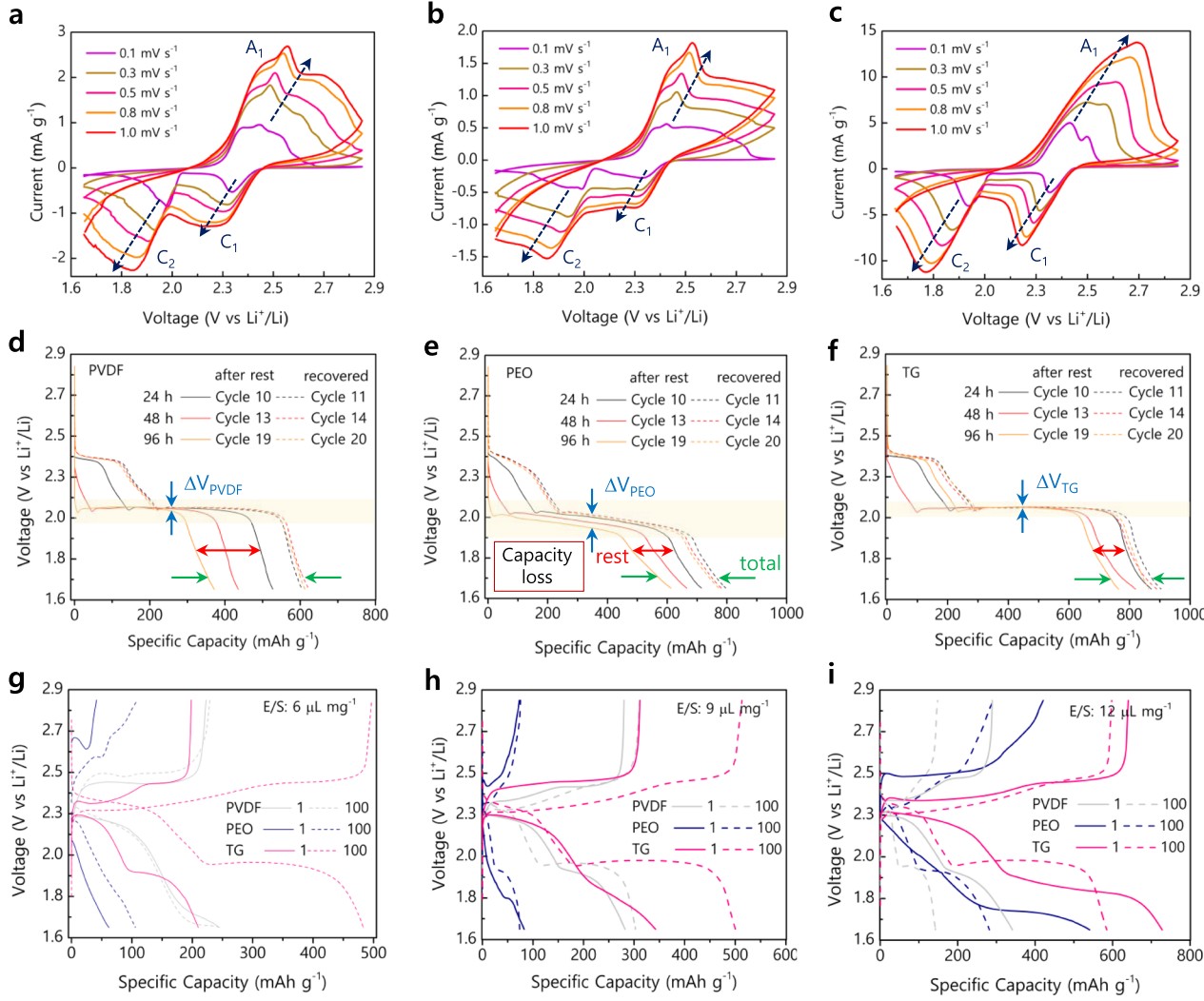

**Fig. 4 Electrochemical kinetic, self-discharge, and electrolyte/sulfur ratio studies for S/PVDF, S/PEO, and S/TG cathodes. a–c** Cyclic voltammograms at various scan rates of 0.1–1 mV s$^{-1}$. **d–f** Self-discharge profile shown after resting 24, 48, and 96 h and respective recovery cycle. **g–i** Discharge-charge profile for various electrodes with specified electrolyte/sulfur ratio E/S 6, 9, and 12 μL mg$^{-1}$.

The impact of progressive scan on the current response was validated through sequential cyclic voltammetry scans between 0.1 and 1 mV s$^{-1}$ (Fig. 4a–c). The voltammogram of sulfur electrodes using TG binder (Fig. 4c) showed more pronounced redox peaks exposing high currents and low polarization than S/PVDF and S/PEO electrodes (Fig. 4a, b). Upon subsequent increase in the scan rates from 0.1 to 1 mV s$^{-1}$, all the electrodes witnessed a gradual rise in the current intensities. For instance, at 1 mV s$^{-1}$, the S/TG electrodes exposed a higher current of 15 mA g$^{-1}$, whereas only 2 and 3 mA g$^{-1}$ was notable for S/PEO and S/PVDF electrodes. In addition, the polarization (ΔV) of the redox (C1, C2, and A1) peaks at various scan rates (Fig. 4a–c) validated minimal polarization of S/TG electrode (Supplementary Fig. 7), which would primarily benefit for the diffusion and movement of ions across the binder-electrode/electrolyte interface. Further, the perspective towards thermodynamic origin was validated through galvanostatic intermittent titration technique (GITT) experiments performed at a current rate of C/10 (Supplementary Fig. 7d–f). Considering the entire GITT discharge processes, the sulfur reduction and utilization of S/TG was 80.3% (1346 mAh g$^{-1}$) against 61% (1031 mAh g$^{-1}$) for S/PVDF and 63% (1059 mAh g$^{-1}$) for S/PEO electrodes denoting that the TG binder utilized about 19.3% and 17.3%

more sulfur than the PVDF and PEO binders and higher than the reported aqueous-processable polymers (Supplementary Table 2).

The internal polysulfide shuttle upon resting Li-S cells were examined through self-discharge studies (Fig. 4d–f and Supplementary Figs. 8, 9) at low (0.2 C) and high (1 C) current rates. The discharge profile depicted in Fig. 2d–f exposes minimal ΔV$_x$ and limited discharge capacity losses after resting, and recovery cycle for S/TG than the S/PVDF and S/PEO electrodes, even open-circuit potential (OCP) remained high for S/TG electrodes (Supplementary Fig. 8d). The Coulombic efficiency (CE) based on the respective discharge/charge of the nth cycle ($QD_n/QC_n$) (Supplementary Fig. 8e, f) after resting remained steady for the S/TG electrode at 97.3% and 88.8% after 6 and 96 h, respectively, against 88.7% and 55.6% for S/PVDF and 93.3% and 82.4% for S/PEO electrodes. At a high current rate (Supplementary Fig. 9), the S/TG electrodes exhibited a reasonable CE of 83.3% after 48 h resting. Since the (ir-)recoverable capacity upon resting depends on the reaction of migrated soluble LiPS forming $Li_2S_2/Li_2S$ precipitate in the Li anode[30], the better recoverable capacity of S/TG electrodes is owed to good polysulfide anchoring of TG. Apart, the E/S studies for S/TG electrodes (2.7 mg cm$^{-2}$) at 6 and 9 μL mg$^{-1}$ (Fig. 4g–i and Supplementary Fig. 10) exhibited specific capacities of 249 and 342 mAh g$^{-1}$, respectively, for the

1st cycle. As the cycles progressed, the electrode underwent sustained wettability to deliver increased capacities of 478 (191%) and 500 mAh g$^{-1}$ (146%) at the end of the 100th cycle. Under a similar E/S ratio, the S/PVDF and S/PEO electrodes delivered poor capacities of 280 and 82 mAh g$^{-1}$, respectively; however, increasing the E/S 12 µL mg$^{-1}$ allowed the electrodes to achieve better capacity but at the expense of 41.64% and 51.47% capacity decay. Conversely, at an E/S 12 µL mg$^{-1}$, the S/TG cathode features high capacity retention of 98.48% (Supplementary Fig. 10f–i). Moreover, lowering E/S causes an increase in polarization due to the nucleation and growth of Li$_2$S leading to sluggish kinetics[31], the S/TG electrodes too follow. Meanwhile, even at a low to marginal E/S ratio, the S/TG offers improved cyclability and capacity retention due to good electrode wettability (Supplementary Table 3). Thus, the S/TG's enhanced electrochemical property is due to well-accessible sulfur particles enabled by the polysaccharidic backbone of the TG binder. These polysaccharides possess abundant polar functional groups like hydroxyl (–OH), cyclic oxygen (–O–), carboxyl (–COOH), and amine (–NH) groups that afford effective polysulfide trapping and overcome its dissolution to maintain the electrolyte viscosity. Meanwhile, the partial ability of PVDF and PEO in trapping polysulfides resulted in its dissolution, which eventually increased the viscosity of electrolytes, leading to a rise in cell resistance[32]. Thus, these results fully support the higher sulfur reduction and utilization of TG achievable as the binder-sulfur interface possessed a lower energy barrier feasible for the Li-ions to permeate and react with the sulfur to a greater extent.

**DFT-guided chemisorption ability**. Apart from higher sulfur reduction and utilization, the capability to trap and reverse lithium polysulfide (Li$_2$S$_n$, $n$ varies by state of charge (SOC)) well favors for stable capacity of the TG binder. The representative discharge profile depicted in Fig. 5a portrays the reduction of sulfur to lithium sulfides as a function of voltage (vs. Li$^+$/Li) and specific capacity$_{(sulfur)}$. Density functional theory (DFT) calculations were conducted to reveal the chemisorption ability of the TG towards high- (Li$_2$S$_n$, $4 \leq n \leq 8$) and low-order (Li$_2$S$_n$, $n < 4$) polysulfides (Fig. 5b, c and Supplementary Fig. 11)[33]. DFT calculations exposed that saccharidic units of TG such as xylose, arabinose, fucose, galactose, and hydroxyproline exert good chemisorption ability towards lithium polysulfides (Fig. 5c). In particular, the adsorption energy towards Li$_2$S$_4$ was found to be higher at −1.14, −1.08, −1.17, −0.83, and −1.21 eV for the saccharidic unit's xylose, arabinose, fucose, galactose, and hydroxyproline, respectively, which is stronger than the interactions between carbons and lithium polysulfides[17] (Fig. 5c and Supplementary Table 4). Apart, the saccharidic units show good adsorption with the low-order Li$_2$S (Fig. 5c) which is the final product during the discharge process. It should be noted from the DFT calculations in accordance with the galvanostatic discharge profile (Fig. 5a) that the TG binder acts precisely in chemisorbing Li$_2$S$_4$ and Li$_2$S through its abundant saccharidic units which largely limits the dissolution and precipitation of LiPS into the liquid electrolyte. Thus, the TG binder's chemisorption towards polysulfides mainly benefits the Li-S cells in two ways, (1) maintaining the concentration of liquid electrolyte during the series of electrochemical conversion, i.e., from solid-state S$_8$ to soluble Li$_2$S$_4$ and to Li$_2$S precipitate, and (2) minimizing the accumulation of residual "dead" sulfur through Li$_2$S$_4$ dissolution and Li$_2$S precipitation (Supplementary Fig. 2). Furthermore, the highly branched TG biopolymer inherits several polar functional groups, such as –OH, –O–, –COOH, and –NH groups of polysaccharides capable of acting as a Lewis base enduring strong adsorption to form a bond with lithium of lithium polysulfides[34].

Likely, the abundant adsorption centers, i.e., polar groups inherited in polysaccharides, traps polysulfides to a greater extent and allow their deposition around the centers to draw forth low resistance across the sulfur-binder/electrolyte interface as notable from the electrochemical property of the S/TG electrode.

The excellent chemisorption ability of the TG binder was revealed through theoretical predictions, and the nature of the solid-electrolyte interface (SEI) layer was experimentally verified through X-ray photoelectron spectroscopy measurements performed for pristine and cycled electrodes (Fig. 5d–l and Supplementary Fig. 12). Upon cycling, the S2$p$ spectra exposed three additional peaks at 167.1, 169.3, and 170.6 eV corresponding to the formation of thiosulfate (166.1–168 eV) and polythionate (168–172 eV) complex, whereas the peak respective of the terminal and bridging sulfur was significantly diminished owing to increased sulfur utilization and reduction[35,36] (Fig. 5e, h). However, a minor portion of the peak at 163.0 eV (Fig. 5h) corresponds to the Li polysulfide (Li$_2$S$_x$), which is an ingredient of the SEI layer[37]. Apart, the C1$s$ spectra (Fig. 5g) expose additional peaks than pristine electrode (Fig. 5d), which are owed to the polycarbonate CO$_3^{2-}$ (290.3 eV) and C-F (293.6 eV) species reflecting the composition of SEI layer formed through the electrochemical decomposition of LiTFSI and ether groups of the electrolyte[38]. In addition, the C–F (–CF$_3$) group is also evidenced in the F1$s$ spectra (Fig. 5j) through a peak at 689.1 eV along with the Li-F peak at ~685.2 eV. Thus, the formation of Li-F traces is beneficial during the prolonged cycles as they suppress Li dendrites by preventing the contact of Li metal with electrolyte, which would lead to improved efficiency and safety of the Li-S cells[38,39]. Further, the N1$s$ spectrum for cycled electrode displayed an additional peak at 398.8 eV apart from a pyridinic-N peak at 400.3 eV due to hydroxyproline (Fig. 5f, i), where the former one is ascribed to the interaction of the amine group with the lithium polysulfide[40]. Further, the formation of lithium bonds was revealed through the Li1$s$ (Fig. 5k) and O1$s$ (Fig. 5l) spectra depicting two peaks at 55.6 eV and 56.4 eV and 531.2 eV, respectively, indicating strong evidence for the Li-S and Li-O bond formation as a result of the inductive effect of electronegative S and O atoms, which is in accordance with theoretically predicted lithium bond formation with polar groups[37,41] (Fig. 5k). The XPS results in conjunction with DFT predictions confirm the strong polysulfide adsorption capability of TG, one of the positive attributes for improved specific capacity apart from higher sulfur reduction and utilization.

**Structure-electrode properties of the binder**. The excellent adhesion and mechanical stability of the S/TG electrode are owed to the polar functional groups of the TG binder, which were first substantiated through Fourier-transform infrared spectroscopy (FT-IR) (Fig. 6a). For the TG binder, the broadband at 3435 cm$^{-1}$ is assigned to the O–H and N–H groups and a peak at 2968 cm$^{-1}$ denotes C–H stretching from the carboxyl group. Several other peaks at 1626–1740, 1415, and 937 cm$^{-1}$ also represent stretching C=O, in-plane C–O–H bonding, out-of-plane O–H groups, respectively, denoting the stretching vibrations of methylene and carbonyl, carboxylate groups. The absorption bands between 1250 and 1035 cm$^{-1}$ denote the stretching vibrations of polyols, ethers, and alcohol groups[42]. These functional groups bonded to carbon atoms were further evidenced through the NMR spectra depicted in Fig. 6b. For more, the mechanical properties of the electrodes were quantitatively evaluated through peeling and nanoindentation measurements (Fig. 6c–f). The adhesion test performed through the electrode patterning showed good adhesion of S/C particles to the Al current collector in S/TG electrodes, and only a minimal portion is detached upon peeling,

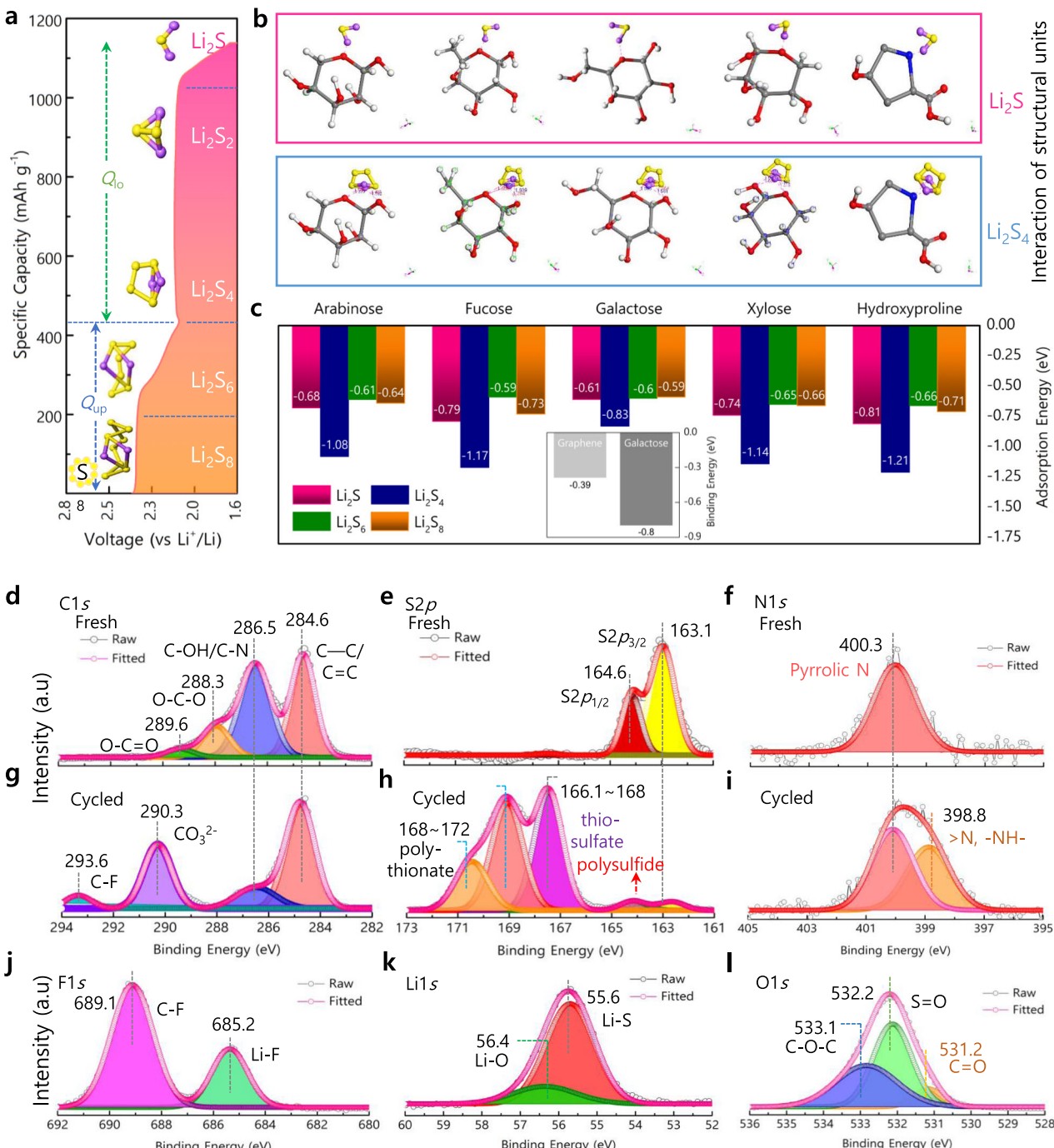

**Fig. 5 Simulation and XPS studies elucidating the adsorption capability of TG binder. a** Discharge profile of S/TG electrode. **b**, **c** DFT analyses; **b** Interaction of chemical units arabinose, fucose, galactose, xylose, and hydroxyproline with Li$_2$S and Li$_2$S$_4$. **c** Calculated absorption energy between units of TG with Li$_2$S$_8$, Li$_2$S$_6$, Li$_2$S$_4$, and Li$_2$S compared with carbons (shown as inset). **d–l** XPS analyses; **d–f** Spectra of C, S, and N for fresh S/TG cathodes. **g–l** Spectra of C, S, N, F, Li, and O post-cycling of S/TG electrodes.

whereas the S/PVDF and S/PEO electrodes showed more detachment (Fig. 6c). The strong adhesion of S/TG particles with the Al current collector is due to the highly branched polar/nonpolar functional groups of TG via van der Waals forces. Nanoindentation studies for S/TG electrodes obtained in a dry-state (no electrolyte) as portrayed in Fig. 6d possessed marginal indentation depth ($h_{max}$) at peak load and final depth ($h_f$) and revealed the highest elastic modulus (Fig. 6e) and hardness (Fig. 6f). It should be noted that the retention of the electrode's mechanical property in a wet-state (with electrolyte, i.e., inside a

battery) is more important to maintain the electrode architecture. Thus, such an elastic feature and hardness observed through the nanoindentation test could have contributed to providing excellent structural and mechanical resilience to the S/TG electrodes during the volume changes (16%, see a section electrode architecture upon cycling in the Supplementary) while maintaining the interparticle contact (Supplementary Note 1 and Supplementary Fig. 13) even at high current rates. Moreover, the strong binding force and hardness imparted from TG to the S/TG cathode facilitated the endurance limit of the binder to exceed internal

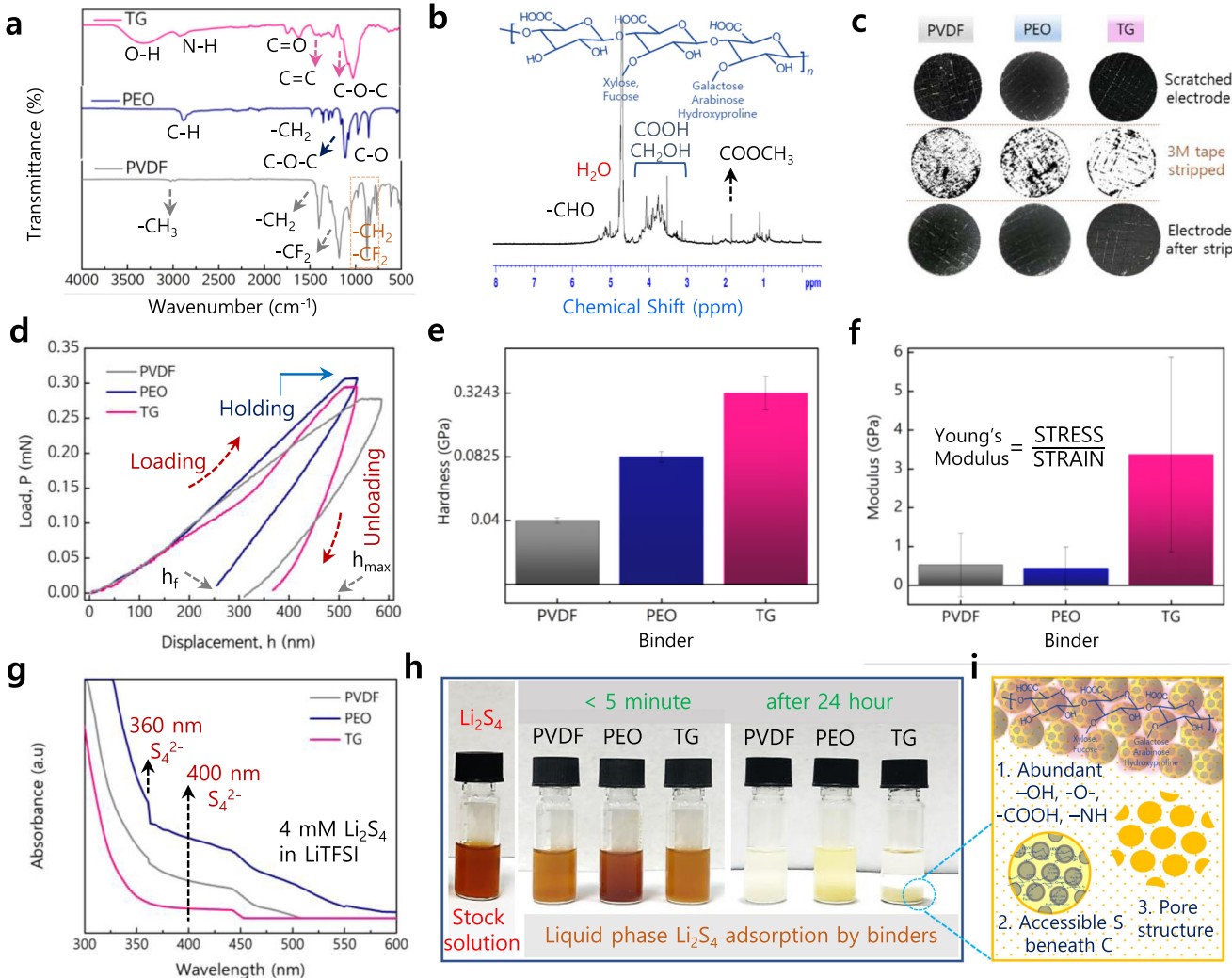

**Fig. 6 Spectroscopic and mechanical studies for S/PVDF, S/PEO, and S/TG electrodes. a** FT-IR spectra for various sulfur electrodes. **b** 1H NMR spectrum for pure TG. **c** Adhesion and peeling-off test. **d** Nanoindentation hardness test. **e** Hardness. **f** Elastic moduli. **g** Adsorption studies through UV-Visible. **h** Static adsorption of liquid phase polysulfides by pristine PVDF, PEO, and TG binders. **i** Factors for high polysulfide adsorption of TG binder.

stresses created during the high-volume expansions leading to an intact S/TG electrode architecture.

Furthermore, electrochemical and theoretical predictions suggesting excellent polysulfide adsorption for the TG were experimentally validated through ultraviolet-visible (UV-Vis) spectroscopic measurements (Fig. 6g). An equal amount of PVDF, PEO, and TG binders were steeped into $Li_2S_4$ catholyte solution, and the respective color changes were observed. After 24 h of immersion, the catholyte solution containing TG turned colorless and evidenced no absorbance around 360 nm in UV-Vis spectra (Fig. 6g, h). Meanwhile, a pale and dark yellow suspension was observable for the solutions containing PVDF and PEO binders (Fig. 6h). The good absorptivity of polysulfides dispersed in the electrolyte by TG binder is ascribed to the chemisorption ability of the abundant polar functional groups and well-controlled electrode microstructures (Fig. 6i).

To explain in great detail the benefits of TG as a binder, the improved electrochemical and mechanical property of sulfur cathodes employing TG is attributable to the highly branched polysaccharide backbones of the binder. Macromolecular *tragacanth* forms structure containing mixtures of α-L-arabinofuranose and 1–4-linked β-D-galactopyranose backbone to an acidic poly-1-4-linked α-D-galacturonate complex[43,44]. The complex and branched polysaccharide backbone is inherited five

significant units, such as D-galactose, D-glucuronic acid, L-arabinose, fucose, and hydroxyproline imparting a surface anionic charge. Owed to their abundant polar functional groups, such as –OH, –O–, –COOH, and –NH, the biopolymer imparts good adhesion of electrode constituents to the current collector, interparticle contacts, and mechanical integrity to the electrode. Predominantly, the swellable nature of TG upon hydrolysis forms a gel (gelation increases with time), i.e., while making electrode slurry, where the highly branched saccharides form links between the chain units to evolve as larger polymers. Specifically, bassorin (tragacanthic) and tragacanthin (arabinogalactan) are two major water-swellable fractions; bassorin exposes a rod-like molecular structure composed of 1,4-linked d-galactose residues with side chains of d-xylose units attached to the main chain by 1,3 linkage. Apart, tragacanthin forms a highly branched arabinogalactan adopting a spherical shape that is probably composed of 1,6- and 1,3-linked d-galactose units attached chains of 1,2-, 1,3-, and 1,5-linked l-arabinose[45,46]. Collectively, the colloidal macromolecular gel affords maximum space for the dispersion of sulfur particles and good interparticle contact (sulfur and carbon), making them entirely covered with binder molecules. Besides, the rod-like and spherical molecular conformations of TG fractions[43–46] ensure a stretchable and conformal framework, which is not only beneficial for three-dimensional access of Li-ions to cause

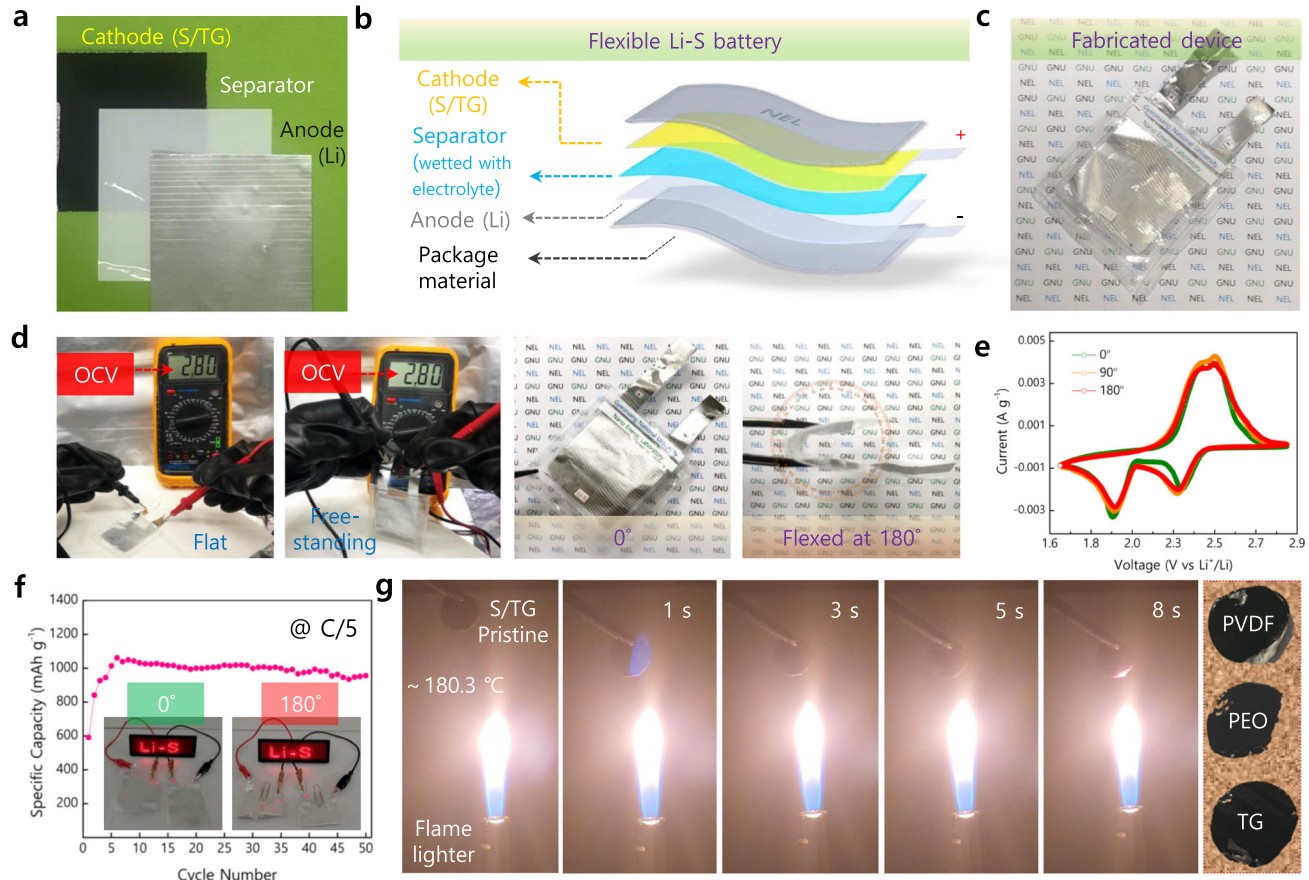

**Fig. 7 Fabrication and performance of flexible Li-S (FLS) battery. a** Digital photograph of the inner configuration of Li-S full cell. **b** Schematic illustration of device fabrication. **c** Digital photograph of assembled FLS device. **d** Photograph displaying open-circuit voltage (OCV) of FLS device under flat, free-standing, and bending angles. **e** Cyclability of FLS device shown through CV tests performed at 0.5 mV s$^{-1}$ in flat (0°) and flexing angles (90° and 180°). **f** Cycling performance at a current rate of C/5, inset photograph depicts the illumination of several light-emitting diodes powered by FLS device. **g** Flame-retardant ability of S/TG electrodes at various exposure times and the recovered electrodes after the flame test.

efficient and high sulfur reactivity (Supplementary Fig. 14) but also act as buffer spaces to offer good elasticity and mechanical integrity to the electrodes during the volume changes. Such architecture of TG also benefits in trapping the polysulfides at the molecular level through their abundant polar functional groups (Fig. 6i).

**Real-time performance towards compact and flexible Li-S battery.** The real-time applications of S/TG electrodes in terms of commercial Li-S batteries that prerequisite high energy and power density, shape conformal (flexible), and safety concerns were demonstrated (Fig. 7). For this purpose, flexible Li-S batteries (FLS) were fabricated with S/TG as cathode, polypropylene separator (Celgard 2500), Li metal as an anode to form a stack which is infiltrated with LiTFSI in DME: DOL (2% LiNO$_3$) electrolyte. Figure 7a, b depicts the specific configuration of components (4 × 4 cm in dimension) and the overall schematics of the FLS device. As fabricated FLS device exhibited a robust and flexible nature while the open-circuit voltage (OCV) remained stable at 2.80 V at different flexing conditions such as flat, free-standing and bending states (0° and 180°) (Fig. 7c, d). Representative voltammograms obtained at a scan rate of 0.5 mV s$^{-1}$ at different flexing angles (0°, 90°, and 180°) exhibited similar (Fig. 7e) redox nature, denoting the FLS device possessed good shape conformal and mechanical strength capable of operating at rough conditions.

Furthermore, the FLS device subjected to galvanostatic cycling at a current rate of 0.2 C delivered a stable and reversible capacity of 982 mAh g$^{-1}$ at the end of the 50th cycle (Fig. 7f), and a gravimetric and volumetric energy density of 243 Wh kg$^{-1}$ and 230 Wh L$^{-1}$ (Supplementary Table 5) was realized at an optimal sulfur mass loading of 3.84 mg cm$^{-2}$ with an E/S of 8 μL mg$^{-1}$. Moreover, a single FLS device could glow several light-emitting diodes (inset of Fig. 7f) at 0° and 180° positions. Apart from the exquisite properties of TG, such as high sulfur utility and reduction, polysulfide adsorption, good adhesion, and mechanical strength, they also endow excellent flame-retardant property and good thermal stability. The flame-retardant property of sulfur electrodes was demonstrated by directly exposing the electrodes to flame and allowed until ignited. Electrodes fabricated with PEO ignited immediately upon exposure to flame and burnt vigorously, whereas S/PVDF electrodes started to burn around 4 s (Supplementary Fig. 15). Interestingly, the S/TG electrodes, upon contact with the flame, suppressed and self-extinguished the flame and kept thermally stable even upon exposure to about 8 s (Fig. 7g). For comparison, the recovered electrodes showed maximum deterioration for S/PEO and S/PVDF electrodes; strikingly, the S/TG electrode retained its shape (distortion at the curved face observable at the bottom of S/TG is folded inward) along with good adhesion of electrode particles against the surface detachment in S/PVDF and S/PEO electrodes (Fig. 7g). The apparent flame-retardant property of the TG binder stems from the formation of the crosslinked thermostable polymer

upon removing water, which might act as an insulating barrier between the heat and active-electrode particles (Supplementary Figs. 16, 17 and mechanism discussed in Supplementary Note 2).

## Discussion

In summary, our study reported an aqueous-processable biopolymer for sulfur cathodes, which exerts an essential positive prerequisite for high-performance and safer practical lithium-sulfur battery. Sulfur electrodes fabricated using TG polymer featured higher sulfur reduction and utilization of 73.9%, accounting for a high initial specific capacity of 1239 mAh g$^{-1}$ at 0.2 C and even good cycling characteristics at 4 C and high sulfur loadings. Increased sulfur utilization proportionate to the volume change and polysulfide generation was meticulously controlled through the crosslinked and highly branched rod-like and spherical molecular conformation of bassorin, tragacanthin, and saccharidic binder and its inherited polar groups as supported through DFT calculations. The exceptional elasticity confirmed through nanoindentations kept the electrode's volume changes within 16%, even at higher current rates. A flexible Li-S device with an ideal mass loading could deliver an energy density of 243 Wh kg$^{-1}$ even in harsh conditions. Such a compact energy storage device and flame-retardant sulfur cathodes epitomize a significant step toward realizing a practical high-performance flexible and safer Li-S battery. Furthermore, the aqueous-processable electrode preparation ensures a green fabrication approach.

## Methods

**Materials**. LiTFSI (99.95%), anhydrous DOL (99.8%), anhydrous DME (99.5%), NMP (99%), super P carbon, and tragacanth were purchased from Sigma–Aldrich. LiNO$_3$ (99.999%) and PEO were purchased from Alfa Aesar. The sulfur powder was obtained from Duksan ultrahigh pure chemicals, and battery-grade PVDF powder and Al foil were procured from MTI corporation. High purity Li foil was obtained from Lithsun Corporation.

**Characterization**. Chemical compositions and the functional groups present in the biopolymer were analyzed using Fourier-transform infrared (FT-IR, Perkin Elmer UATR2) spectrometer in the spectral range (4000–500) cm$^{-1}$ and $^1$H nuclear magnetic resonance (NMR, Bruker Advance-300) spectrometer. The surface morphology and cross-sectional profiles of pristine and cycled electrodes were observed under a field-emission scanning electron microscope (FESEM, TESCAN LYRA$_3$). Energy-dispersive X-ray spectroscopy (EDS) spectra and elemental maps were obtained on a Tescan LYRA$_3$ scanning electron microscope equipped with an Oxford X-MaxN EDS detector. The valence states and chemical bonding of the binder elements in pristine and cycled electrodes were verified through X-ray photoelectron spectrometer (XPS, Thermo Scientific NEXSA) equipped with Al Kα (hν = 1486.6 eV) radiation with 50 eV of pass energy, the raw data were interpreted through ADVANTAGE software provided. Deconvolution of XPS peaks and fittings were performed through Origin 9.0 graphing and analysis software. Mechanical properties such as hardness and Young's modulus (Elastic modulus) of the electrodes were performed through nanoindentation tests (Anton Parr, MHT[3]) with 90° conical indenter at multipoint, and the values were calculated using the Oliver-Pharr method[47]. Ultraviolet-visible spectroscopy (UV-Vis, SCINCO Neosys-2000) was employed to study the adsorption capability of binders after 24 h of steeping, the measured spectral ranges from 300 to 600 nm. The thermal stability of the binders was studied through thermogravimetric analysis (TGA, SCINCO N-1000) performed at 10 °C min$^{-1}$ between 30 and 900 °C in N$_2$ atmosphere.

**Preparation of sulfur electrodes with various binders**. Initially, mixtures containing sulfur and super P carbon in the mass ratio 7:3 were prepared by gentle mixing and were subjected to melt-infiltration (sulfur infused to carbon) at 155 °C for 12 h in N$_2$ atmosphere (100 mL min$^{-1}$) to form sulfur/carbon (S/C) composite. Typically, fabrication of working electrodes involved intermixing the respective ratio 80:15:5 of S/C composite as a sulfur host, MWCNT as a conductive material, and either of the binder (PVDF, PEO, or TG) was made into a viscous slurry using solvents (N-methyl-2-Pyrrolidone for PVDF, C$_2$H$_5$OH/H$_2$O (7:3) mixture for PEO and ultrahigh pure aqua for TG). The resultant viscous slurry was homogeneously cast on an aluminum foil (17 μm thickness) through the doctor blade technique and was naturally dried at room temperature for 12 h and later vacuum dried at 80 °C for 8 h to ensure complete removal of solvents. The dried electrodes were calendared and punched into round disks of φ12 mm to serve as the working electrode. The active material loading was around 1.1–1.4 mg cm$^{-2}$; however, the

various loading masses of 2.4, 4.7, and 8.1 mg cm$^{-2}$ were fabricated for mass variation studies.

**Electrochemical characterization**. Electrochemical performance for sulfur electrodes fabricated using various binders was studied using CR2032 type coin cells assembled inside an argon-filled glove box maintained at O$_2$ and H$_2$O levels below 0.1 ppm. Typically, the coin cell stack consisted either of sulfur electrodes (PVDF, PEO, or TG) as a cathode (working electrode), monolayer polypropylene membrane (Celgard 2500) as a separator, and pure Li disk as an anode (reference or counter electrode). The stack was wetted with a controlled amount, 30–35 μL of an in-house prepared liquid electrolyte, which comprised of 1 M lithium bis(trifluoromethane sulfonyl)imide (LiTFSI) salt dissolved in a eutectic anhydrous mixture (1:1, v/v) of 1,3 dioxolane (DOL) and 1,2-dimethoxymethane (DME) with 2% lithium nitrate (LiNO$_3$) as an additive to ensure smooth SEI formation. The cells were crimped and then directly used to evaluate the electrochemical properties of the working electrodes. All the electrochemical measurements involving cyclic voltammetry (CV), galvanostatic charge-discharge (GCD), and galvanostatic intermittent titration technique (GITT) tests were conducted on an automatic multichannel battery cycler (WonATech, WBCS3000L32). CV studies were performed between the potential window 1.7–2.8 V (vs. Li$^+$/Li) at various selected scan rates 0.1–1.0 mV s$^{-1}$. GCD tests were performed at the various current rate (C-rates) determined from the theoretical capacity of sulfur (1 C = 1675 mA g$^{-1}$), and the reported specific capacity values are calculated based on the active mass of sulfur present in the cathode. GITT tests for the cells were performed from OCV to lower cut-off voltage 1.7 V at C/10 with a 5 min interval for discharge or charge and a minute delay. Electrochemical impedance spectroscopy (EIS) tests were carried out at open-circuit voltage in the frequency range 1 MHz–0.01 Hz with an AC amplitude of 5 mV through a potentiostat/galvanostat (WonATech, ZIVE SP).

**DFT calculation**. The optimized atomic configurations and adsorption energy between the polysaccharide binder and lithium polysulfides (Li$_2$S$_n$, 8 ≤ n ≥ 1) were calculated through the Density Functional Theory (DFT) approach. Electron exchange correlations were defined within the Perdew-Burke-Ernzerhof (PBE) Generalized Gradient Approximation (GGA)[48] implemented in the Dmol3 package of Materials Studio 7.0 (Accelrys Inc.). Self-consistent field (SCF) calculations were performed until the tolerance attained below 10$^{-6}$ eV atom$^{-1}$. The adsorption energy was calculated for the five major saccharidic molecule models in the tragacanth gum, such as arabinose, fucose, xylose, fructose, and hydroxyproline. The adsorption energy ($E_{ads}$) was calculated from the energy differences between adsorbed lithium polysulfide on the saccharidic surface ($E_{total}$) and pristine saccharide ($E_{pristine}$), which is defined as $E_{ads} = E_{total} - E_{pristine} - E_{LiPS}$, where $E_{LiPS}$ is the energy of lithium polysulfide.

**Adhesion test**. Adhesion experiments were performed on a sample size φ12 mm preferably in the electrode dimension. Calendared circular electrodes were patterned in the form of a grid (1 × 1 mm) drawn using single-edge carbon steel blades, and the resultant electrode dust was gently blown off before the peeling test. For the peeling-off test, Scotch Magic tape was gently affixed to the surface of the patterned electrodes and then peeled off with minimal mechanical force. After the peeling test, the respective loss of mass was calculated from the difference in pristine patterned and after peeling electrodes. Digital photographs of the electrodes pre(post)-peeling test and peeled off Scotch Magic tapes were considered for visualizing the adhesion capability of materials to the Al current collector.

**Fabrication of flexible Li-S device**. Flexible lithium-sulfur (FLS) battery was fabricated in 4 × 4 cm (l × b) dimension constituting of typical configuration S/TG working electrode, monolayer polypropylene separator (Celgard 2500), and Li foil as an anode housed in polyethylene amide (PEA) package material. Initially, S/TG electrode was prepared as detailed earlier (refer section: preparation of sulfur electrodes with various binders), pure Li foil of optimal mass ratio was stacked against S/TG with polypropylene separator placed between the electrodes to form working component, and the electrodes were connected to the external leads from outside and finally encased within two PEA films through hot pressing. The stack was wetted with an electrolytic solution (~8 μL mg$^{-1}$) right before the final casing; the used electrolyte was 1 M LiTFSI dissolved DOL: DME with 2% LiNO$_3$ as an additive.

**Flame-retardant property**. Pre-weighed calendared electrodes of φ12 mm were directly exposed to the flame emanant tip from the flame lighter. The flame-retardant property of the electrodes was estimated based on the time that the electrodes were exposed to the flame and remained until it started to burn, while the flame is supplied continuously during the process. The time recorded to self-extinguish to the burning of electrodes was validated for exposing the safety of the sulfur cathodes.

## Data availability

The data that support the findings of this study are available within this paper/supplementary information, any additional data are available from the corresponding author upon reasonable request.

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

## Acknowledgements

This research was supported by the National Research Foundation of Korea (NRF) funded by the Ministry of Education (2021R1I1A3A04034294) and the Ministry of Science and ICT (2021R1A4A1030318).

## Author contributions

C.S. and H.Y.J. conceived the idea, designed the experiments, and analyzed the data. C.S. performed experimental, theoretical and electrochemical studies. S.S.K. involved in the characterization. H.Y.J. supervised the research work. All the authors discussed the results and contributed to the manuscript.

## Competing interests

The authors declare competing interests in this work as the invention described is the subject of a patent application (KR 10-2021-0100309) on July 30, 2021.

**Additional information**

