## [Peer Review File · Nature Communications]

Flame retardant high-power Li-S flexible batteries enabled by bio-macromolecular binder integrating conformal fractionsREVIEWER COMMENTS

Reviewer #1 (Remarks to the Author):

The work by Senthil et al. describes a new binder for sulfur cathodes in Li-S batteries. The comparison of this binder with two others is carried out with impressive detail and, while I have some criticisms discussed in detail below, I am convinced that there is a significant improvement over the controls. However, I find the manuscript difficult to read and I am not yet convinced there is a significant improvement compared to other functional binders that warrant publication in Nature Communications. I may be convinced after the authors address my comments below:

The introduction is very poorly written and almost not understandable. It is more than an English language correction and requires significant restructuring. In addition, the entire results and discussion section is too dense and difficult to read. I suggest a significant rewrite focusing on the most important results and leaving some of the supporting analysis to the SI. The work is quite thoroughly investigated – which is great – but it makes the paper difficult to read...

Why is the activation process so much more extreme in the TG case?

In Figure 3d, there is no indication of what rates these electrodes are cycled at nor any mention of the rate performance for these, more interesting, high loading cases. Somehow they seem to be cycling better than the low loading case in 3a-c. Why? Longer cycling in comparison to controls should be carried out to better highlight the impact of this work.

The diffusivity of Li ions was estimated from cyclic voltammetry using the Randles-Sevcik equation which is derived with assumption that almost certainly do not apply here (flat electrode surface, diffusion limited). I suggest the authors do not use this quantitatively but discuss the CV results qualitatively.

It is unclear why GITT was used in the way it was – to estimate the capacity of the sulfur. Normally, this would be used to gain more information like the diffusivity for some limiting conditions that don't hold here (your particle size is likely too small for GITT analysis to be valid). This just adds length and distracts from your story.

The authors discuss that increased hardness and modulus is responsible for the improved stability. However, wouldn't this just lead to a more brittle fracture? I would imagine that other mechanical properties, not studied here, are more relevant to buffering the expansion and contraction like the maximum elastic strain or elongation at break, for example.

For the DFT calculated interaction energies, the values are somewhat meaningless unless compared to values obtained for other polysulfide scavengers used in the literature.

Error bars for the thickness estimates from SEM?

Where is the "dead sulfur" or residual sulfur you are looking at in the EDX in the SI? This is not mentioned. How can one distinguish this from sulfur impregnated into the cathodes in the SEM imaging?

The performance improvements over existing literature in Supplemental Table S1-S2 cannot provide an accurate comparison and are difficult to read. I would suggest using a comparison which considers as many aspects as possible. For example, areal capacity improvements might come at the expense of thick or heavy electrodes. For Table S2, the capacity decay rate is emphasized but for which loadings? It is easy to get low decay rates with low loading cathodes.

Related to my previous comment, why do the authors choose to report volumetric capacity when this is not a particularly interesting metric for a Li-S battery which isn't expected to outcompete more dense Li-ion batteries. Gravimetric energy density is more important and should be estimate instead considering the masses of all components (maybe not the actual electrolyte amount since this must be made higher due to problems with the anode that you didn't circumvent in this work – something that might be mentioned somewhere in the paper).

Reviewer #2 (Remarks to the Author):

This manuscript reports a novel biopolymer as the binder for sulfur cathodes in Li-S battery. Authors conducted a comprehensive study through experimental investigations and theoretical studies. The developed binder endows the battery remarkable electrochemical performance. The manuscript

provides enough data, and the discussion is also reasonable. The work is of sufficient quality, and potential significance. I recommend this work to be published in "Nature Communication" after addressing following issues:

1. "Nature Communication" publishes high-quality papers that report important research findings. Therefore, authors need to elaborate the importance of this work, such as new concept or new mechanism, etc. Only good performance is not enough.
2. Super P is a well-known conductive material for constructing working electrodes for Li-S battery. Why MWCNT was applied as the conductive material in this work? Is there any specific reason?
3. In supplementary Table 1, authors made a comparison of various reported binders in Li-S batteries. For the column of "rate", the unit was specified as "mA cm⁻²". However, the units of some listed values are "C", including the value provided in this work. It's better to use the same unit to make the comparison more clear. The same thing happens in Supplementary Table 2.
4. In the market, the energy density of Li-S battery has reached more than 400 Wh kg⁻¹. In some reported research, Li-S pouch cell achieved a practical specific energy of over 300 W h kg⁻¹. This work reported a stack energy density of 230 Wh L⁻¹. How to compare these values? What if the unit "Wh L⁻¹" in this work is converted into "W h kg⁻¹"?
5. There are a few grammatical mistakes.

Reviewer #3 (Remarks to the Author):

This manuscript reports a new plant binder TG, which obviously enhanced the cycling stability of elemental sulfur cathode, compared to organic binder PVDF and PEO. Major revisions are required as the following,

Actually, there are lots of reports indicating that aqueous binder, such as, PAA, LA 132, modified β -Cyclodextrins, guar gum and so on, demonstrate better performances than traditional PVDF binder for elemental sulfur cathodes. If possible, please compare TG with aqueous binders.

High loading is a good result. What was the tap density of elemental sulfur cathode with TG binder?

E/S ratio is an important parameter for elemental sulfur cathode. Does TG binder reduce E/S ratio remarkably?

How about the self-discharge property of elemental sulfur cathode with TG binder?

It is difficult to understand TG working as flame retardant for elemental sulfur. Please further explain.

Response to Reviewers Comments (Manuscript NCOMMS-21-27788-T)

We thank and are incredibly grateful to the reviewers for providing valuable suggestions to improve further and strengthen the manuscript's impact. As suggested by the reviewers, all the comments raised were carefully considered and addressed in the revised manuscript. Necessary experiments have been performed, and the results were included as five figures and four tables along with discussions to elucidate the impact of macromolecular tragacanth gum as a biopolymeric binder for high-performance and flame-retardant Li-S batteries, while in-sights on the mechanism for the high sulfur utility, self-discharge property, electrolyte/sulfur ratio studies, and flame-retardant ability of S/TG cathodes were provided. The corresponding inclusions in the revised manuscript are highlighted through track changes. The experiments, results, and discussions performed for the revision are briefly summarized as follows.

1. Electrochemical performances and cycling stability studies of high mass loaded S/PVDF and S/PEO cathode controls have been tested and compared with S/TG electrodes.
2. Effect of electrolyte/sulfur (E/S) ratio on the overall Li-S cell performance was performed for all the S/TG, S/PVDF, and S/PEO cathodes through galvanostatic charge/discharge studies and comparatively investigated.
3. Electrode thickness studies for the cycled S/PVDF, S/PEO, and S/TG cathodes have been conducted through SEM and are compared.
4. Tap density of S/PVDF, S/PEO, and S/TG cathodes have been conducted.
5. Electrode wettability tests have been performed to estimate the electrolyte uptake and the effect of the E/S ratio on the capacity for all the electrodes.
6. Self-discharge studies for all the S/TG, S/PVDF, and S/PEO electrodes were performed through electrochemical cycling at two different rates, high (1 C) and low (0.2 C) current rate with respective resting times patterns as 6 h – 12h – 24 h – 48 h – 72 h – 96 h (total resting 10.8 days).

We have made appropriate changes according to the reviewers' suggestions, as detailed in the following paragraphs. We believe that incorporating these changes has greatly enriched the quality of this manuscript.

REVIEWER COMMENTS

Reviewer #1:

The work by Senthil et al. describes a new binder for sulfur cathodes in Li-S batteries. The comparison of this binder with two others is carried out with impressive detail and, while I have some criticisms discussed in detail below, I am convinced that there is a significant improvement over the controls. However, I find the manuscript difficult to read and I am not yet convinced there is a significant improvement compared to other functional binders that warrant publication in Nature Communications. I may be convinced after the authors address my comments below:

Author reply: *The authors express their sincere thanks to the reviewer for valuable suggestions to further improve the scientific content and the manuscript's impact. The authors have revised the manuscript as per the reviewer's suggestion and have re-written the introduction, results, and discussion part at necessary instances that conveys the scientific story in a simple and better way; also, the authors regret the difficulty caused by a few instances. Necessary experiments were undertaken, and the respective results and discussions were included in the revised manuscript and supplementary file. The corresponding inclusions were highlighted through track changes in the revised manuscript, and for clarity, we have included the graphs and their explanation alongside the specific comment.*

Comment 1: The introduction is very poorly written and almost not understandable. It is more than an English language correction and requires significant restructuring. In addition, the entire results and discussion section is too dense and difficult to read. I suggest a significant rewrite focusing on the most important results and leaving some of the supporting analysis to the SI. The work is quite thoroughly investigated – which is great – but it makes the paper difficult to read...

Author reply: *The authors regret making the manuscript tricky, along with dense content in a few instances. As per the reviewer's suggestion, the authors have taken utmost importance to make the revised manuscript much easier to read through simple scientific language. Also, as pointed by the reviewer, few of the experimental electrochemical results corresponding to CV and GITT studies were shifted to the supporting information to have a smooth flow of the manuscript content and style. The authors are glad about the reviewer's suggestion on the thorough investigation of this study.*

Comment 2: Why is the activation process so much more extreme in the TG case?

Author reply: *The authors thank the reviewer for an interesting observation. As pointed out by the reviewer, the authors agree that the activation process in S/TG electrodes is at par higher than the S/PVDF and S/PEO electrodes. Taking cognizance from the experimental understanding obtained through electrochemical impedance spectroscopy, the solution resistance exerted by the S/TG electrode is intermediate within the values obtained for S/PEO and S/PVDF electrodes. It is well documented that the propagation of initial discharge or reduction process occurring at all the electrodes undergo with the partial electrode's wettability even though conditions such as sufficient amount of electrolyte and appropriate resting time is employed. Thus, the reactivity of the electrodes during the initial process is dependent on the electrode's wettability, where the capability of electrodes to attain higher wettability primarily allows for more reactivity of lithium with the sulfur active particles. To validate this claim, we have performed a wettability test for all the electrodes by soaking them in an electrolytic solution composed of 1 M LiTFSI in DOL: DME with 2% LiNO₃ over a period ranging from initial soaking – 24 h. From the results, it was found that the S/TG electrodes exhibited higher wettability at 211.94 %, which is higher than the S/PVDF and S/PEO electrodes after 24 h of steeping in an electrolytic solution. However, the S/PVDF and S/PEO electrodes show comparatively lower wettability of 117.6 and 144.92 % after 24 h. Such an observation from the wettability results exposes that the S/TG electrodes are capable of wetting faster and higher to electrolytes that ought to provide access to Li⁺ ions causing high sulfur utility. Such a high wettability of electrode knocks for the higher sulfur utility of 73.9 % for S/TG electrodes, driving a high activation process as the proportion of sulfur utilized is much higher during the initial cycle.*

Secondly, the galvanostatic charge/discharge studies performed at various current rates are shown in Fig. 3, elucidating the high reactivity of sulfur particles for S/TG electrodes. As evident from the results, the sulfur utility at the initial cycle was calculated as 73.9% for S/TG electrodes, whereas the S/PEO and S/PVDF electrodes were capable of undergoing sulfur reactivity at 66.2% and 48.5%, respectively, at a current rate of 0.2 C. The difference in the sulfur reactivity of S/PEO and S/PVDF electrodes comparing S/TG electrodes were 7.7% and 25.4%, respectively, which conveys that the S/TG electrodes could cause higher utility of sulfur. Such a higher utility of sulfur in S/TG electrodes causes more activation, especially during the initial cycle. In contrast, the S/PVDF and S/PEO electrodes require more electrolytes or possess an unfavorable chemical environment to cause better electrode wettability to allow for more sulfur activity in their electrodes.

ACTION: On page 7 in the revised manuscript, "Apart from the initial cycle, where the reduction peaks are shifted due to dominant activation processes as owed to high sulfur reactivity, the subsequent scans show overlapped cathodic (C1 and C2) and anodic (A1) peaks indicative of excellent cycling stability in the S/TG electrode. On the contrary, the S/PVDF and S/PEO electrodes still showed a gradual diminish and shift in the redox profile as the scans proceeded due to their decreased access to sulfur particles."

Comment 3: In Figure 3d, there is no indication of what rates these electrodes are cycled at nor any mention of the rate performance for these, more interesting, high loading cases. Somehow they seem to be cycling better than the low loading case in 3a-c. Why? Longer cycling in comparison to controls should be carried out to better highlight the impact of this work.

Author reply: *The authors thank the reviewer for the comment. The current rates followed for the electrochemical performance of high mass loaded electrodes 2.4 mg cm⁻², 4.7 mg cm⁻², and 8.1 mg cm⁻² are 0.1 C (1 C = 1.675 A g⁻¹).*

The S/TG electrodes exhibited higher electrochemical performances than S/PVDF and S/PEO electrodes. Especially the reversible capacity at all the mass loadings was found to be better in S/TG electrodes, as noted by the reviewer. The high mass loaded S/TG electrodes shown in Fig. 3d delivered good reversible capacities of 3.82, 6.3, and 10.02 mAh cm⁻² at the end of the 100th cycle, which is comparable with low mass loaded electrodes shown in Fig. 3(a-c). It is known that the increase in the areal loadings of electrodes causes dissimilar physico-mechanical properties than the low areal loadings. Thus, in order to have a good electrode architecture as shown in Supplementary Fig. 6, we have typically followed a new method of fabricating high mass loaded electrodes termed as "gelation" in a way such that they possess similar electrode architecture as observed for the lower mass loaded ones. In this way, the concerns regarding the high interparticle resistances, ion movement, and volume changes in the electrodes could be minimized to achieve high electrochemical performances. It is worth mentioning that the use of macromolecular TG as a binder for sulfur cathodes has its own benefits in terms of the electrochemical activity and the other concerns mentioned above when compared with the organic PVDF and PEO counterparts.

- *First, the TG binder's property to dissolve in an aqueous medium makes it amicable in a way that the formation of slurry desired for the fabrication of electrodes could be altered to a wide*

range of viscosity to form a "swellable-gel" by just prolonging the duration of slurry preparation, which we referred as "gelation". Referring to the molecular structure of TG binder, bassorin and tragacanthin are two major water-soluble polymeric fractions that intrinsically possess the property of swelling when they are exposed to an aqueous medium for a longer duration. Thus, the swellability of these fractions enables to accommodate high loading of the electrode constituents. Therefore, the swelling nature, i.e., in other ways, could be represented as volume expansion of the binder in an aqueous medium is made use of to engulf a high quantity of the electrode constituents and allow them to be fixed upon gentle drying. Through this process, the electrodes exposed suitable physical and mechanical property as notable from Supplementary Fig. 6. As a result, the high-loaded electrodes exhibit good electrochemical properties. It is worth mentioning that the process of gelation does not require an additional binder or aqueous medium, while an increase in the time of slurry formation enables us to achieve a gel-like viscous slurry.

- Secondly, (a) the macromolecular TG binder possesses a complex and branched polysaccharidic backbone that contains abundant polar functional groups. These polar functional groups such as -OH, -O-, -COOH, and -NH involve in chemically adsorbing the polysulfides leading to decreased polysulfides as also predicted through the DFT approach. Interestingly, through these macromolecular polysaccharidic groups, as formed polysulfides are well controlled in both cases of low and high mass loaded electrodes. (b) The S/TG electrodes possess a good electrode architecture, an essential criterion for high mass loaded electrodes which is verified through the adhesion and peel-off test notable from Fig. 6c. The good adhesion and intact electrode architecture of S/TG enable good ion and electron movement leading to good sulfur reactivity and utility. (c) Through the "gelation" strategy, the S/TG electrodes, incredibly high mass loaded electrodes retain the electrode architecture achieved for low mass loaded ones. Specifically, the sulfur reactivity and utility, movement of ions and electrons, and mitigating the volume change through the conformably available rod and sphere fractions of TG binder boost the high mass loaded electrodes to perform comparably with the low mass loaded electrodes. Collectively, properties such as physically robust and chemically viable for the sulfur redox reaction favor the better electrochemical performance of high mass loaded electrodes.

As pointed by the reviewer, the comparison of mass loading with S/PVDF and S/PEO electrodes were performed and included as Fig. 3f,g in the revised manuscript, and the respective explanations are provided in the revised manuscript pages 12-13.

ACTION: On page 12 in the revised manuscript, "The resultant method allowed the loading mass of sulfur to successfully exceed even above 12 mg cm^{-2} with a tap density of S/TG powders at 1.08 g cm^{-3} . The S/TG electrodes of 2.4 mg cm^{-2} , 4.7 mg cm^{-2} , and 8.1 mg cm^{-2} were evaluated at a current rate of 0.1 C (Fig. 3d),"

On pages 12-13 in the revised manuscript, "Besides, the robust nature of high sulfur-loaded TG electrodes is prominent from the long-term cycling stability shown in Fig. 3f,g. The S/TG cathodes tested with sulfur loading 2.4 mg cm^{-2} exhibited an areal capacity of 2.4 mAh cm^{-2} and demonstrated excellent cycling stability featuring 95.2% capacity retention even after 500 cycles. Under similar conditions, the S/PVDF and S/PEO electrodes of mass 2.5 and 2.2 mg cm^{-2} exhibited discharge capacities of 0.17 and 1.18 mAh cm^{-2} , respectively, with severe capacity fade. The case is more for high sulfur-loaded S/PVDF and S/PEO electrodes, while S/TG showed a marginal decay. This observation could be applied to develop robust and high active mass loading sulfur cathodes as Li-ion diffusion and transport are influenced by the electrode thickness."

In Figure 3f,

In Figure 3g,

Comment 4: The diffusivity of Li ions was estimated from cyclic voltammetry using the Randles-Sevcik equation which is derived with assumption that almost certainly do not apply here (flat electrode surface, diffusion limited). I suggest the authors do not use this quantitatively but discuss the CV results qualitatively.

Author reply: *The authors sincerely thank the reviewer for the comment and agree with the reviewer's suggestion on the applicability of cyclic voltammetry. The intent of the authors to use cyclic voltammetry was to reveal a few valuable electrochemical information during the initial cycles, which are related to peak potential separation (ΔV_{1-3}) and the peak current deviation ($i_{p,a}/i_{p,c}$) for the TG binder comparing PVDF and PEO. It is known that the potential and concentration in a reversible electrochemical system follow the Nernst equation and/or derived ones (the current i is proportional to the square of scan rate $v^{1/2}$). As sulfur cathodes undergo redox reaction rather than complete electrode-adsorption, the authors employed the Randles-Sevcik equation to describe the peak current i_p and determine the diffusion properties. However, upon the reviewer's opinion, the experimental results as Figures related to sequential CV at various scan rates were shifted to supplementary information, and the text parts were entirely removed in the revised manuscript.*

ACTION: As suggested by the reviewer, Fig. 4d-f in the manuscript corresponding to diffusion studies through CV measurements were shifted to the revised supplementary information as supplementary Fig. 7a-c. Further, the explanation related to diffusional properties of all the S/PVDF, S/PEO, and S/TG electrodes were limited to the discussions related to redox peak, peak current, peak shift, polarization, and reversibility.

Comment 5: It is unclear why GITT was used in the way it was – to estimate the capacity of the sulfur. Normally, this would be used to gain more information like the diffusivity for some limiting conditions that don't hold here (your particle size is likely too small for GITT analysis to be valid). This just adds length and distracts from your story.

Author reply: *The authors agree with the reviewer's comment on GITT being used as a source to obtain information related to the diffusivity of ions in the electrodes. Since the biopolymer tragacanth is the first report to be used as a binder for sulfur cathodes in Li-S batteries, the authors felt that the need for thermodynamic and kinetic electrochemical analyses is essential. Thus, the elucidation of the electrochemical properties of S/TG electrodes compared to the S/PVDF and S/PEO electrodes required understanding from two different aspects, namely through static evidence and dynamic perspectives. Firstly, the static evidence for the S/TG cathodes in Li-S cells was revealed through the electrochemical impedance spectroscopy, as notable from Fig. 1e. Although the resistances in the S/TG cells are minimal, more profound insights on the thermodynamic aspects of how the TG binder influences the sulfur redox kinetics, concentration of polysulfides at different potential and reactions in sulfur cathodes required a dynamic perspective, which was revealed through electrochemical tools such as cyclic voltammetry and galvanostatic intermittent titration technique. Based on the reviewer's opinion, the figures corresponding to GITT results (Fig. 4g-i) were shifted to the revised supplementary information as supplementary Fig. 7d-f.*

ACTION: As suggested by the reviewer, Fig. 4g-i in the manuscript related to GITT measurements were shifted to the revised supplementary information as supplementary Fig. 7d-f and the discussion part related to GITT studies were limited to have a smooth flow of the scientific content.

Comment 6: The authors discuss that increased hardness and modulus is responsible for the improved stability. However, wouldn't this just lead to a more brittle fracture? I would imagine that other mechanical properties, not studied here, are more relevant to buffering the expansion and contraction like the maximum elastic strain or elongation at break, for example.

Author reply: *The authors thank the reviewer for an interesting comment. The authors have investigated the mechanical properties of the electrodes through the nanoindentation, adhesion, and peel-off tests, which revealed high elastic modulus, hardness, and adhesion for the S/TG cathodes than*

the S/PVDF and S/PEO controls. Based on the results, high elastic modulus and hardness are responsible for excellent structural and mechanical resilience to the S/TG electrodes during cycling. Considering the electrode architecture and engineering, the authors would like to clarify that the electrode's mechano-chemical property in a secondary battery is crucial in two different conditions. (i) Dry state after electrode fabrication and (ii) Wet state during the electrochemical cycling in a battery. It is well known that the conventional Li-ion batteries employ polyvinylidene difluoride (PVDF) as a standard binder for the fabrication of electrodes because of its inertness both chemically and electrochemically during the battery cycling. Considering the above criteria, PVDF possesses low elastic modulus and hardness where the electrodes ought to remain un-fragile during fabrication and while battery operation. On the contrary, the PVDF electrodes (i.e., any active material + any conductive carbon + PVDF) bind with the electrode particles weakly, exposing low elastic modulus. However, PVDF's elastic modulus is degraded in a wet state (battery containing electrolyte), promoting mechanical failure of the electrodes, such as crack, delamination, and deterioration upon cycling, which is even evident in our study with S/PVDF electrodes (Supplementary Fig. 13 a1-a3) and the literature. Significantly, the situation is worse when active materials such as silicon or sulfur-based anodes are used as they are prone to severe volume changes (>300% for silicon and >80% for sulfur) upon cycling. To validate this claim, post-mortem SEM analysis depicted in Supplementary Fig. 13 concluded that the S/PVDF electrodes exposed cracks or brittle within electrode at multiple points leading to electrode deterioration reflected as severe capacity decay. On the other side, S/PEO electrodes with intermediate mechanical properties possessed comparatively lower cracks. In the case of S/TG electrodes, good electrode architecture was maintained after cycling (Supplementary Fig. 13c1-c3), which clarifies that the high Young's moduli and hardness due to strong binding forces in the biopolymer tragacanth facilitate the endurance limit of the binder exceeding the internal stresses in the electrodes caused during the high-volume expansions while cycling. Further, the ex-situ SEM postmortem results clarify that despite being low or high Young's moduli material in the dry state, the same condition could not be applied or assumed in the wet state as the real picture is on how does the binder supports the active material and remain during the battery operation. Thus, for the active materials that undergo more volume expansion, binders need to be stiff enough to overcome the internal stress. Hence, it is not only important to consider the physical properties of binders and macro-scale adhesion testing such as elastic modulus and hardness of electrodes in a dry state (after fabrication, W/O electrolytes) (Adv. Mater. 25, 1571-1576 (2016)), but also significant importance is required in a wet state (filled with electrolyte),

i.e., during the operation of batteries as we have elucidated through the ex-situ post-mortem SEM analysis (Supplementary Fig. 13) of all the electrodes.

From the perspective of Li-S cell chemistry, binder particles (affirming the other electrode particles) wetted in a liquid electrolyte offer a form of nano-scale connections between the heterogeneous electrode particles and metallic current collectors. Thus, it is realistic to assume that a nanometer-level thin layer of binder molecules in battery electrodes exhibits different chemical, electrochemical, and mechanical behaviors upon contact with an electrolytic solution under variable electric potential, concentration gradient, and temperature when compared with bulk and dry polymers. As a result, interparticle affinity and mechano-chemical properties between the binder and the other constituents of electrodes may significantly impact the mechanical integrity and electrochemical stabilities of the electrode. As documented in several studies (J. Power Sources, 112, 61–66, (2002), J. Power Sources, 189, 72–80, (2009), Adv. Mater., 25, 1571–1576, (2013)), it is understood that the adhesive force of the binder with other electrode constituents and retention of Young's modulus in the electrolyte are critical factors that determine the suitability of the binder. Thus, sulfur being active material that undergoes severe volume changes, the electrode S/TG need to be strong enough to withstand the stress produced during the Li-S interconversion reaction. This phenomenon is due to the strong binding forces, and higher stiffness of binders facilitate the endurance limit of the binder exceed the internal stresses in the electrodes caused during the high-volume expansions, which is why we observe a limited expansion of the S/TG electrodes at 17% against 22% and 37% as shown in Supplementary Fig. 13d.

ACTION: Supplementary Figure 13,

Comment 7: For the DFT calculated interaction energies, the values are somewhat meaningless unless compared to values obtained for other polysulfide scavengers used in the literature.

Author reply: Following the reviewer's suggestions, the interaction energies calculated for the polysulfides and saccharic interaction were compared with literature and tabulated as Supplementary table 4 in the revised supplementary information and Fig. 5c in the revised manuscript.

ACTION: Comparison of DFT analysis results as Supplementary table 4 and Fig. 5c were included in the revised supplementary information and the revised manuscript.

Supplementary Table 4, "Comparison of binding energy values obtained through DFT analyses for several reported metals, metalloid, semi-metal, and polymeric binders with the TG binder in Li-S batteries."

Class of Material	Bonding type	Binding energy (eV)	Electrode material (Sulfur+Host or Additive)	Capacity after n th cycle	Ref.
Non-metal	S-C	-1.64–2.84	C/S host	860 (500)	31
	S-O	-0.84–1.95	O-CNTs/S host	800 (600)	32
	Li ₂ S _x -N	-0.25–4.08	N-doped C/Graphene	752 (300)	33
			g-C ₃ N ₄ /S composite	620 (500)	34
Li ₂ S _x -P	-0.94–1.39	P-doped CNTs,	917 (100)	35	
		N, P-doped graphene	638 (500)	36	
Metalloid	Li ₂ S _x -B	-0.9–5.15	V-doped C coated CNTs/S	562 (500)	37
			N, B-doped C/S	556 (500)	38
Metal	S-Ti	-2.33	S in porous C/TiO ₂	800 (200)	39
			Ti ₄ O ₇ /S	800 (250)	40
	Li ₂ S _x -V	-1.9	V ₂ O ₃ /C	921 (100)	41
			V ₂ O ₅ /C	900 (100)	42
	S-Mn	-1.84–2.59	MnO ₂ /S	802 (300)	43
			S coated MnO ₂ sheets	1030 (200)	44
	S-Fe	-0.84–1.85	FeS ₂ in S/C	700 (200)	45
			Fe ₂ O ₃ in S	575 (100)	46
	S-Co	-3.79	CoS ₂ in Graphene/S	750 (250)	47
			Co ₃ O ₄ -S	656 (200)	48
	S-Ni	-0.72–2.59	S/NiS@C spheres	718 (200)	49
nanocrystal S@NiS ₂			954 (1200)	50	
S-Cu	-1.56	S/Cu	1300 (80)	51	
		S/C aerogel with CuS	1073 (100)	52	
Polymer	Li-F	-0.4	PVDF	~510 (100)	53
	O-Li/N-Li	-1.81/	PPA	~740 (400)	54
	Li-O	-1.14	PVP	760 (100)	52
	Li-O	-0.52	PVA	515 (250)	21
	Li-O	-	CMC	~390 (350)	57
	Li-N	-1.24	PEI	~930 (100)	55
	Li-N	-0.59	PANI	628 (200)	56,57
	O-Li/S-Li	-3.59/-2.47	DSM	686 (350)	58
	Li-O	-2.17	APP	640 (400)	59
	O-Li/N-Li	-1.17/-1.21	Tragacanth	976 (200)	Our work

Figure 5c in the revised manuscript,

On page 15, "In particular, the adsorption energy towards Li₂S₄ was found to be higher at -1.14 eV, -1.08 eV, -1.17 eV, -0.83 eV, and -1.21 eV for the saccharidic unit's xylose, arabinose, fucose, galactose, and hydroxyproline, respectively, which is stronger than the interactions between carbons and lithium polysulfides (Fig. 5c and Supplementary Table 4)."

Comment 8: Error bars for the thickness estimates from SEM?

Author reply: The authors thank the reviewer for the comment on thickness estimates as it is essential to reveal the volume change characteristics of electrodes post-cycling. Cross-section thickness was measured for the fresh and cycled electrodes at multi-points per sample and several cycled electrode batches. The error observed for S/PVDF, S/PEO, and S/TG electrodes are 35±0.02 μm, 35±0.05 μm, and 35±0.05 μm for fresh electrodes and 48±0.03 μm, 44±0.04 μm, and 42±0.02 μm for cycled electrodes, respectively.

ACTION: As per the reviewer's suggestion, the error bars based on the multi-point analysis and other batch samples were now included. The results were provided as a bar graph in supplementary Fig. 13e-f and as supplementary table 8.

Supplementary Figure 13e-f,

Supplementary Table 8, “SEM thickness estimates for fresh and cycled S/PVDF, S/PEO, and S/TG electrodes.”

Test	S/C: MWCNT: PVDF		S/C: MWCNT: PEO		S/C: MWCNT: TG	
	thickness (μm)	Error	thickness (μm)	Error	thickness (μm)	Error
Fresh						
A	34.95	0.056	36.00	0.016	36.00	0.05
B	35.00	0.006	36.10	0.084	36.10	0.05
C	35.05	0.044	36.10	0.084	36.00	0.05
D	35.01	0.004	36.00	0.016	36.10	0.05
E	35.02	0.014	35.90	0.116	36.00	0.05
F	35.01	0.004	36.00	0.016	36.10	0.05
Average	35.006	0.0213	36.016	0.0553	36.05	0.05

Test	S/C: MWCNT: PVDF		S/C: MWCNT: PEO		S/C: MWCNT: TG	
	thickness (μm)	Error	thickness (μm)	Error	thickness (μm)	Error
Cycled						
A	48.00	0.016	44.00	0.058	42.00	0.016
B	48.05	0.034	44.00	0.058	42.10	0.084
C	48.00	0.016	44.05	0.008	42.00	0.016
D	48.00	0.016	44.10	0.042	42.00	0.016
E	47.95	0.066	44.10	0.042	42.00	0.016
F	48.10	0.084	44.10	0.042	42.00	0.016
Average	48.016	0.0386	44.058	0.0416	42.016	0.0273

Comment 9: Where is the "dead sulfur" or residual sulfur you are looking at in the EDX in the SI? This is not mentioned. How can one distinguish this from sulfur impregnated into the cathodes in the SEM imaging?

Author reply: *The authors thank the reviewer for the insightful observation. The authors have attempted to elucidate the loss of sulfur species (after cycling) from the electrodes by comparing cycled electrodes with fresh ones. As it is known that the electrochemical reaction of S and Li proceeds through the multi-steps producing high-order and low-order polysulfide that are soluble in an electrolyte (Li_2S_4) and the insoluble end product (Li_2S), which are considered lost sulfur from the electrodes. Thus, an attempt was made to have a clue from this perspective; however, due to the limited reliability on the quantitative data of low atomic number elements through EDS-SEM, the results may not be defined appropriately. Thus, the authors have removed the corresponding explanation of residual sulfur related to EDS-SEM from the manuscript.*

Comment 10: The performance improvements over existing literature in Supplemental Table S1-S2 cannot provide an accurate comparison and are difficult to read. I would suggest using a comparison which considers as many aspects as possible. For example, areal capacity improvements might come at the expense of thick or heavy electrodes. For Table S2, the capacity decay rate is emphasized but for which loadings? It is easy to get low decay rates with low loading cathodes.

Author reply: *The authors thank the reviewer for the suggestion to have a more detailed comparison. Following the reviewer's comments, Table S1 and S2 in the supplementary information has been updated with maximum details under four section headings cell parameter, cell performance, organic and aqueous-based binders compared with reported binders to that of our TG binder. The revised tables S1 and S2 were provided in the revised supplementary information.*

ACTION: Table S1 and S2 updated with details under two sections, namely cell parameter and cell performance, were included in the revised supplementary information, which is provided below.

In Supplementary Table 1, "Comparison of sulfur mass loading for variously reported binders in Li-S batteries (C-rate based on sulfur's theoretical capacity 1.675 A g^{-1})."

Binder (B)	Cell parameter				Cell performance						Ref.
	Electrode ratio (%) S/C/B	Sulfur load (mg cm ⁻²)	E/S ratio (μL mg ⁻¹)	Rate (C)	Areal Capacity (mAh cm ⁻²)		Cycle no.	Capacity retention (%)	Current collector		
					Initial	Final					
PVDF	7:2:1	2.0	-	0.4	1.8	1.62	50	90	Ni foam	6	
Gelatin	6.3:3:0.7	2.0	-	0.4	2.3	0.8	50	34	Al foil	7	
PAMAC	8:1:1	3.0	-	0.5	3.2	2.0	105	62.5	C-coated Al foil	8	
DCP	6.4:2.6:0.1	9.8	5	0.5	9.2	6.5	50	70.6	Ni foam	9	
CDp-N+	7:2:1	5.5	7	0.3	6.6	4.4	45	66.6	Al foil	10	
MPEII	5:4:1	6.5	-	0.05	7.52	5	70	66.4	Al foil	11	
PAMAM	8:1:1	4.0	40	0.05	4.56	2.8	100	61.4	-	12	
S/GG-XG	8:1:1	6.5	21	0.8	4.2	~4.7	90	101	-	13	
Carrageenan	-	5-10	-	0.01	~9	~5.8	100	64.4	-	14	
CMC-CA	8.5:4.5:5.5	10	4	1.0	~11.2	~5.7	50	50.8	Carbon paper	15	
PEI	6:3:1	8.6	16	0.05	9.7	6.4	50	65.9	Al foil	16	
Tragacanth gum	8:1.5:0.5	8.1 4.7	12- 15	0.1 0.1	10.8 9.7	10.02 6.3	100	92.7 64.9	Al foil Al foil	Our Work	

In Supplementary Table 2, “Comparison of Li-S battery performance and its stability using various bio-based and conventional binders (C-rate based on sulfur’s theoretical capacity 1.675 A g⁻¹).”

Type of polymer binder	Binder (B)	Cell parameter			Cell performance				Ref.	
		Electrode ratio (%) S/C/B	Sulfur load (mg cm ⁻²)	E/S ratio (μL mg ⁻¹)	Rate (C)	Specific capacity (mAh g ⁻¹)		Cycle no.		Capacity decay per cycle
						Initial	Final			
Organic	PVDF	7:2:1	2.4	50	0.5	788	322	100	0.49	17
	PAA	6:3:1	1.5	-	0.2	390	298	100	0.09	18
	PAA/ PEDOT:PSS	7:2:1	0.8	-	0.5	1121	830	80	0.36	19
	PVA	6:3:1	3.5	9	0.5	936	563	75	0.49	20
Bio-based	Gum arabic	8:2	1.5	100	0.2	1157	841	500	0.054	21
	Guar/ Xanthum	8:1:1	6.5	21	0.5	1000	724	150	0.18	10
	Na-alginate	5:4:1	1.5	-	0.5	776	508	50	0.53	22
	Gelatin	6.3:3:0.7	2.0	-	0.5	950	680	100	0.28	8
	Zein/CNF	6.38:2.41: 0.21	1.8	-	0.6	890	590	500	0.06	23
	SBR: CMC	5.4:3.6:1	1	-	0.2	867	612	150	0.55	24
	Chitosan	6.3:3:0.7	1-1.5	-	0.2	1145	780	100	0.36	25
	Gelatin/ Chitosan	6.3:3:0.7	1-1.5	-	0.5	857	675	200	0.09	27
	Soy protein- PAA	6.75:2.25: 0.1	1.5	10	0.3	790	595	200	0.12	26
	LA-132	7:2:1	2	80	0.5	901	470	100	0.43	27
	LA-133	5.4:3.6:1	1	-	0.2	1176	709	150	0.31	22
	Catechol- Chitosan sulfate	8.5:0.5:1	1.3- 1.5	15	0.2	785	605	400	0.045	28
	Laponite	8:1:1	1	-	0.5	810	595	500	0.05	29
	Polyurethane -Phytic acid	-	0.8- 1.5	25	0.5	1051	632	500	0.08	30
	Cyclodextrin -S composite	7:2:1	2.5	7	0.5	692	636	50	0.02	AFM
	Tragacanth gum	8:1.5:0.5	1.1- 1.4	15-17	0.2 1	1239 1236	976 865	200 300	0.13 0.12	Our work

Comment 11: Related to my previous comment, why do the authors choose to report volumetric capacity when this is not a particularly interesting metric for a Li-S battery which

isn't expected to outcompete more dense Li-ion batteries. Gravimetric energy density is more important and should be estimate instead considering the masses of all components (maybe not the actual electrolyte amount since this must be made higher due to problems with the anode that you didn't circumvent in this work – something that might be mentioned somewhere in the paper).

Author reply: *The authors thank the reviewer for an interesting comment on the practical aspects of Li-S batteries. As per reviewers' suggestions, the authors have taken account to elucidate the values in terms of the mass, i.e., the gravimetric energy density of as-fabricated flexible Li-S device, which is calculated as 243 Wh kg⁻¹.*

ACTION: The gravimetric energy density values were included in the revised manuscript. Further, the details related to the device fabrication, cell parameters, and cell performances were updated on page 39 in the revised supplementary information.

On page 2 in the revised manuscript, “the flexible Li-S battery delivers a stack gravimetric energy density of 243 Wh kg⁻¹,” **On page 21,** “a gravimetric and volumetric energy density of 243 Wh kg⁻¹ and 230 Wh L⁻¹ (Supplementary Table 5) was realized at an optimal sulfur mass loading of 3.84 mg cm⁻² with an E/S of 8 μL mg⁻¹.” **On page 22,** “A flexible Li-S device with an ideal mass loading could deliver energy densities of 243 Wh kg⁻¹”

Reviewer #2:

This manuscript reports a novel biopolymer as the binder for sulfur cathodes in Li-S battery. Authors conducted a comprehensive study through experimental investigations and theoretical studies. The developed binder endows the battery remarkable electrochemical performance. The manuscript provides enough data, and the discussion is also reasonable. The work is of sufficient quality, and potential significance. I recommend this work to be published in "Nature Communication" after addressing following issues:

Author reply: *The authors sincerely thank the reviewer for providing valuable time and comments to further strengthen the content and importance of the manuscript. All the points raised by the reviewer have been carefully considered, and the respective experiments were performed. The results and discussions pertaining to the reviewer's comments have been included in the revised manuscript and supporting information. For clarity, we have included the graphs and explanation alongside the specific comment.*

Comment 1: "Nature Communication" publishes high-quality papers that report important research findings. Therefore, authors need to elaborate the importance of this work, such as new concept or new mechanism, etc. Only good performance is not enough.

Author reply: *The authors sincerely thank the reviewer for a critical suggestion to further elucidate the importance of this work. The importance of the work is detailed as follows.*

A new macromolecular biopolymer tragacanth is proposed for the first time as a multifunctional binder for the sulfur cathodes in Li-S battery. The novel biopolymer fabricated sulfur cathodes, i.e., S/TG have demonstrated a high sulfur reactivity and low polysulfide shuttle to offer good electrochemical performances. Based on the molecular structure of TG, a new mechanism is proposed to curb the shuttle effect and restrain volume changes, which we coined as "conformal fractions". The presence of rod and sphere-like bassorin and tragacanthin molecular fractions allows for an excellent mechanical property to the S/TG electrode, which provides high Li⁺ ion access to sulfur, control shuttle through the studded saccharidic unit, accelerate ion movement and restrict volume changes. Also, we have elucidated the in-depth understanding of how macromolecular biopolymer endows improved electrochemical performance. In this work, we propose two new concepts to the best of our knowledge, (i) relating polysulfide shuttle with the electrochemical profile characteristics and (ii) electrode engineering to achieve high sulfur loaded electrodes.

First, the profile width (ΔP_w) is proposed as a unique concept derived from the experimental electrochemical discharge profile. The ΔP_w allows establishing a positive correlation between the shuttle effect and the discharge capacity to determine the level of capacity fade due to polysulfide losses. Since the shuttle effect is well known detrimental factor in Li-S battery, the ΔP_w is much necessary to electrochemically understand the degree of sulfur loss.

Next, we propose a new concept, "gelation", to fabricate highly-mass loaded sulfur cathodes exclusively using the tragacanth binder. Using the gelation technique, electrodes can be manufactured with a high sulfur content of good electrode porosity. For the low mass loaded sulfur cathodes of $< 4\text{-}5 \text{ mg cm}^{-2}$, a regular slurry preparation is desired. Upon increasing the sulfur loading, the gelation technique is followed where the binder is allowed to swell, forming a gel-like slurry capable of engulfing more sulfur active particles. Through this concept, not only high loaded electrodes are feasible, but also a crack-free electrode could be achieved with good porosity, which is necessary for achieving good electrochemical performances.

ACTION: As pointed by the reviewer, the new concept and mechanism were elaborated.

On pages 8-9 in the revised manuscript, "In addition, the profile width (ΔP_w), a phenomenon exclusively reported in this study, was measured at a discharge voltage of 2.2 V to determine the irreversible losses in terms of capacity as a function of voltage, which remained relatively much shorter and concise for S/TG electrodes. This parameter is necessary to elucidate the shuttle effect that occurs during the charge process as a consequence of previous discharge. The capacity losses for selective discharges ranging from 50 to 200 cycles were within a variable of 8 mAh g^{-1} for S/TG, comparing S/PVDF of 22 mAh g^{-1} and S/PEO electrodes of 47 mAh g^{-1} (Fig. 2g-h). Such a minimal ΔP_w variable for TG is only possible upon efficient trapping and reversibility of polysulfides as the cycle progresses without paving the way for the detrimental shuttle effect that is well known to induce capacity decay as prevalent for either S/PVDF or S/PEO electrodes."

On page 12, "In order to overcome this issue, we propose "gelation" a swellable-gel-like viscous slurry of TG binder was initially prepared by prolonging the duration of slurry formation and used to disperse the active sulfur/carbon mixture through gentle addition (90-120 minutes) and then laminated."

On pages 19-20, "Macromolecular tragacanth forms structure containing mixtures of α -L-arabinofuranose and 1-4-linked β -D-galactopyranose backbone to an acidic poly-1-4-linked α -

D-galacturonate complex. The complex and branched polysaccharide backbone is inherited five significant units such as D-galactose, D-glucuronic acid, L-arabinose, fucose, and hydroxyproline imparting a surface anionic charge. Owing to their abundant polar functional groups such as -OH, -O-, -COOH, and -NH, the biopolymer imparts good adhesion of electrode constituents to the current collector, inter-particle contacts, and mechanical integrity to the electrode. Predominantly, the swellable nature of TG upon hydrolysis forms a gel (gelation increases with time), i.e., while making electrode slurry, where the highly branched saccharides form links between the chain units to evolve as larger polymers. Specifically, bassorin (tragacanthic) and tragacanthin (arabinogalactan) are two major water-swellable fractions; bassorin exposes a rod-like molecular structure composed of 1,4-linked d-galactose residues with side chains of d-xylose units attached to the main chain by 1,3 linkage. Apart, tragacanthin forms a highly branched arabinogalactan adopting a spherical shape that is probably composed of 1,6- and 1,3-linked d-galactose units attached chains of 1,2-, 1,3-, and 1,5-linked l-arabinose. Collectively, the colloidal macromolecular gel affords maximum space for the dispersion of sulfur particles and good interparticle contact (sulfur and carbon), making them entirely covered with binder molecules. Besides, the rod-like and spherical molecular conformations of TG fractions ensure a stretchable and conformal framework, which is not only beneficial for three-dimensional access of Li-ions to cause efficient and high sulfur reactivity (Supplementary Fig. 14) but also act as buffer spaces to offer good elasticity and mechanical integrity to the electrodes during the volume changes."

Comment 2: Super P is a well-known conductive material for constructing working electrodes for Li-S battery. Why MWCNT was applied as the conductive material in this work? Is there any specific reason?

Author reply: *We thank the reviewer for an interesting comment on the use of MWCNT as a conductive carbon for the electrode reported in this study. As pointed out by the reviewer, Super P carbon is a well-known conductive agent employed to fabricate working electrodes in secondary batteries. It is known that the choice of conductive carbon depends on the electronic conductivity of the active material. Li-S battery being a high theoretical capacity contender for high energy density batteries, the critical challenges associated with its operation, as detailed below, enabled the authors to employ MWCNT as a conductive carbon to fabricate working electrodes.*

- (i) *The active material and its end-discharge product, namely sulfur and Li₂S generated according to the electrochemical equation ($S_8 + 16 Li^+ \leftrightarrow 8Li_2S$), are insulative in nature. Particularly, the conductivity of sulfur ranges about $5 \times 10^{-30} \text{ s/cm}^{-1}$, which is very poor to conduct electrons within the cathodes. The unique nanostructure and high electrical conductivity of MWCNT than other carbon materials interested its use as a conductive carbon to improve the overall electronic conductivity of the sulfur cathode (Chem. Commun., 2013, 49, 10545–10562).*
- (ii) *MWCNTs can form a continuous conductive network in a 3D space which accelerates the electronic conductivity, while short electron diffusion distance is possible (J. Mater Chem. A, 2019, 7, 17204-17241).*
- (iii) *Even distributions of the sulfur active particles are among the criteria for better reactivity of sulfur cathodes which is made possible through the MWCNTs, which can hold the active sulfur particles within the abundant macropores. Thus, the space for accommodating more sulfur particles in the electrode is increased upon comparing with other carbon materials (J. Mater. Chem. A, 2016, 4, 775–780).*
- (iv) *Shuttling of the dissolved polysulfides could be controlled to some extent with the use of MWCNTs; thus, the loss of sulfur and low Coulombic efficiency is prevented to ensure reasonable comparison among the controls (J. Electrochem. Soc., 2003, 150, A889–A893).*
- (v) *The volume expansion is yet another challenge for sulfur cathodes. The volume changes during the sulfur-conversion reaction increase to about 80%, making the cathodes fragile and could also block the ion channels upon using the normal conductive carbon materials. The use of MWCNTs could partially alleviate the volume changes as they possess good mechanical strength, thus leading to decreased electrode deterioration and a feasible ion pathway (Adv. Funct. Mater., 2014, 24, 6105–6112).*

Thus, considering the above positive effects, the authors found that the use of MWCNT would be a better choice than the other carbon materials to be employed as conductive carbon for the fabrication of working electrodes.

Comment 3: In supplementary Table 1, authors made a comparison of various reported binders in Li-S batteries. For the column of "rate", the unit was specified as "mA cm⁻²". However, the units of some listed values are "C", including the value provided in this work.

It's better to use the same unit to make the comparison more clear. The same thing happens in Supplementary Table 2.

Author reply: *The authors regret the confusion caused by the current rates "mA/g" and "C" used in supplementary tables 1 & 2. As suggested by the reviewer, the current rates were made uniform to the "C" rate in the revised supplementary tables 1 & 2 to have a good comparison among the reported electrochemical performance values for Li-S batteries.*

ACTION: In Supplementary Table 1,

Binder (B)	Cell parameter				Cell performance						Ref.
	Electrode ratio (%) S/C/B	Sulfur load (mg cm ⁻²)	E/S ratio (μL mg ⁻¹)	Rate (C)	Areal Capacity (mAh cm ⁻²)		Cycle no.	Capacity retention (%)	Current collector		
					Initial	Final					
PVDF	7:2:1	2.0	-	0.4	1.8	1.62	50	90	Ni foam	6	
Gelatin	6.3:3:0.7	2.0	-	0.4	2.3	0.8	50	34	Al foil	7	
PAMAC	8:1:1	3.0	-	0.5	3.2	2.0	105	62.5	C-coated Al foil	8	
DCP	6.4:2.6:0.1	9.8	5	0.5	9.2	6.5	50	70.6	Ni foam	9	
CDp-N+	7:2:1	5.5	7	0.3	6.6	4.4	45	66.6	Al foil	10	
MPEII	5:4:1	6.5	-	0.05	7.52	5	70	66.4	Al foil	11	
PAMAM	8:1:1	4.0	40	0.05	4.56	2.8	100	61.4	-	12	
S/GG-XG	8:1:1	6.5	21	0.8	4.2	~4.7	90	101	-	13	
Carrageenan	-	5-10	-	0.01	~9	~5.8	100	64.4	-	14	
CMC-CA	8.5:4.5:5.5	10	4	1.0	~11.2	~5.7	50	50.8	Carbon paper	15	
PEI	6:3:1	8.6	16	0.05	9.7	6.4	50	65.9	Al foil	16	
Tragacanth gum	8:1.5:0.5	8.1 4.7	12- 15	0.1 0.1	10.8 9.7	10.02 6.3	100	92.7 64.9	Al foil Al foil	Our Work	

In Supplementary Table 2,

Type of polymer binder	Binder (B)	Cell parameter			Cell performance				Ref.	
		Electrode ratio (%) S/C/B	Sulfur load (mg cm ⁻²)	E/S ratio (μL mg ⁻¹)	Rate (C)	Specific capacity (mAh g ⁻¹)		Cycle no.		Capacity decay per cycle
						Initial	Final			
Organic	PVDF	7:2:1	2.4	50	0.5	788	322	100	0.49	17
	PAA	6:3:1	1.5	-	0.2	390	298	100	0.09	18
	PAA/ PEDOT:PSS	7:2:1	0.8	-	0.5	1121	830	80	0.36	19
	PVA	6:3:1	3.5	9	0.5	936	563	75	0.49	20
Bio-based	Gum arabic	8:2	1.5	100	0.2	1157	841	500	0.054	21
	Guar/ Xanthum	8:1:1	6.5	21	0.5	1000	724	150	0.18	10
	Na-alginate	5:4:1	1.5	-	0.5	776	508	50	0.53	22
	Gelatin	6.3:3:0.7	2.0	-	0.5	950	680	100	0.28	8
	Zein/CNF	6.38:2.41: 0.21	1.8	-	0.6	890	590	500	0.06	23
	SBR: CMC	5.4:3.6:1	1	-	0.2	867	612	150	0.55	24
	Chitosan	6.3:3:0.7	1-1.5	-	0.2	1145	780	100	0.36	25
	Gelatin/ Chitosan	6.3:3:0.7	1-1.5	-	0.5	857	675	200	0.09	27
	Soy protein- PAA	6.75:2.25: 0.1	1.5	10	0.3	790	595	200	0.12	26
	LA-132	7:2:1	2	80	0.5	901	470	100	0.43	27
	LA-133	5.4:3.6:1	1	-	0.2	1176	709	150	0.31	22
	Catechol- Chitosan sulfate	8.5:0.5:1	1.3- 1.5	15	0.2	785	605	400	0.045	28
	Laponite	8:1:1	1	-	0.5	810	595	500	0.05	29
	Polyurethane -Phytic acid	-	0.8- 1.5	25	0.5	1051	632	500	0.08	30
	Cyclodextrin -S composite	7:2:1	2.5	7	0.5	692	636	50	0.02	AFM
	Tragacanth gum	8:1.5:0.5	1.1- 1.4	15-17	0.2 1	1239 1236	976 865	200 300	0.13 0.12	Our work

Comment 4: In the market, the energy density of Li-S battery has reached more than 400 Wh kg⁻¹. In some reported research, Li-S pouch cell achieved a practical specific energy of over

300 W h kg⁻¹. This work reported a stack energy density of 230 Wh L⁻¹. How to compare these values? What if the unit "Wh L⁻¹" in this work is converted into "W h kg⁻¹"?

Author reply: *We sincerely thank the reviewer for an interesting comment on the practical aspects of Li-S batteries. As pointed by the reviewer, the energy density of a few Li-S batteries in the market and reports exceeds above 350 Wh kg⁻¹ with a cycle life of over 350 charge/discharge. In light of this, we would first like to highlight our work and then discuss the batteries available in the market.*

Firstly, we demonstrated a flexible pouch cell that employs a novel biopolymer binder tragacanth fabricated S/TG cathodes that achieve reasonable electrochemical performance compared with a lab-scale coin cell. The S/TG electrodes were shown to be capable of withstanding mechanical perturbation at different flexible conditions while delivering significant reduction and oxidation behavior with stable sulfur reactivity. These developments were made with possible lab-scale pouch cell design and engineering as follows.

(i) The S/TG cathodes employed for the flexible pouch cells were single-side coated electrodes that possess good elasticity and electrode integrity notable from the nanoindentation and adhesion and peeling-off test measurements (Fig. 6-7 in the revised manuscript). With this single-side coated electrodes, as-fabricated flexible Li-S cells delivered a reversible capacity of 982 mAh g⁻¹ at a current rate of 0.2 C at the end of 50th charge/discharge cycles, which is depicted in Fig. 7. (ii) Next, our FLS devices were tested in laboratory conditions with a standard electrolyte composed of 1 M lithium bis(trifluoromethane sulfonyl)imide (LiTFSI) salt dissolved in an anhydrous mixture (1:1, v/v) of 1,3 dioxolane (DOL) and 1,2-dimethoxymethane (DME) with 2% lithium nitrate (LiNO₃) as an additive. (iii) In terms of the anode side, the Li foil we have used was fresh with polished Li surfaces, and moreover, the thickness of Li foils was manually reduced to the least possible ones that could be achieved with our laboratory conditions.

With those practical feasibilities, we have demonstrated the use of biopolymer tragacanth binder that has higher edge such as high sulfur reactivity, good polysulfide trap, capacity retention, and mechanical robustness along with flame-retardant property over other binders tabulated in the supplementary table 5 to be applicable in future Li-S batteries.

Now, considering the commercial Li-S batteries available in the market, they are high-tech engineered devices with more function-specific battery components such as (i) two-sided coating of sulfur cathodes with well-engineered electrode constituents (size and shape of sulfur and carbon, thickness of electrode) and their properties, (ii) advanced electrolyte formulations with multiple additives, and (iii)

polymer/ceramic protected unique lithium metal anodes of designated thickness were used to pack Li-S batteries. Thus, the attempt undertaken in this work was with the existing standard conditions available with lab-scale, which may not apply for comparing the market grades as they are engineered in all aspects of cell chemistries. However, as suggested by the reviewer, we will continue to improve the results in terms of practical aspects as the results with the S/TG electrodes are interesting considering the role of the polymer's physical and electrochemical activity. Meanwhile, we believe the successful demonstration of flexible Li-S devices with flame-retardant ability is the next step for flame-retardant high-power Li-S flexible batteries.

As pointed by the reviewer, the gravimetric energy density of the as-fabricated pouch cell was calculated to be 243 Wh kg⁻¹, which was updated in the revised manuscript on page 21. The detailed fabrication process, cell parameters, and cell performances were updated in the revised supplementary information.

ACTION: On page 21 in the revised manuscript, "Furthermore, the FLS device subjected to galvanostatic cycling at a current rate of 0.2 C delivered a stable and reversible capacity of 982 mAh g⁻¹ at the end of the 50th cycle (Fig. 7f), and a gravimetric and volumetric energy density of 243 Wh kg⁻¹ and 230 Wh L⁻¹ (Supplementary Table 5) was realized at an optimal sulfur mass loading of 3.84 mg cm⁻² with an E/S of 8 μL mg⁻¹."

On page 30 in the revised supplementary information,

Parameter for the flexible Li-S battery

	Weight	Mass loading		Size	
Anode	0.1086 g	Anode (Li)	3.0 mg cm ⁻²	Anode	4 × 4
Cathode	0.1428 g	Cathode (S)	3.84 mg cm ⁻²	Cathode	4 × 4
Separator	0.0293 g	Electrolyte/Sulfur ratio	8 μL mg ⁻¹	Separator	4.3 × 4.3
Overall cell		Cell thickness	360 μm		

Total energy

$$\text{Energy (E)} = \text{Capacity (C)} \times \text{Voltage (V)}$$

$$E = 59.31 \text{ mAh} \times 2.24 \text{ V} = 132.85 \text{ mWh}$$

Areal energy (E_a)

$$E_a = \text{Energy (E)} / \text{Area (A)}$$

$$E_a = 132.85 / 16 \text{ cm}^2 = 8.303 \text{ mWh cm}^{-2}$$

Volumetric energy density (E_v)

$$E_v = \text{Areal capacity } (E_a) / \text{Cell thickness}$$

$$E_v = 8.303 / 360 = 230 \text{ Wh L}^{-1}$$

Gravimetric energy density (E_G)

$$E_v = \text{Areal capacity } (E_a) / \text{Cell mass}$$

$$E_v = 8.303 / 341 = 243 \text{ Wh kg}^{-1}$$

Comment 5: There are a few grammatical mistakes.

Author reply: *We thank the reviewer for the suggestion. The authors have thoroughly revised the manuscript to rectify grammatical errors. We once again thank the reviewer for the suggestions which helped to improve our work.*

Reviewer #3:

This manuscript reports a new plant binder TG, which obviously enhanced the cycling stability of elemental sulfur cathode, compared to organic binder PVDF and PEO. Major revisions are required as the following.

Author reply: *The authors extremely thank the reviewer for providing important suggestions to further strengthen the scientific content and impact of this manuscript. Significantly, the comments regarding the tap density, electrode wettability, E/S ratio, and self-discharge properties for the sulfur cathode using TG binder were critical and helpful to advance our work from the practical prospects of Li-S batteries. As suggested by the reviewer, all the experiments were performed, and the respective results as figures and discussion were included in the revised manuscript and revised supplementary files highlighted through track changes. The authors once again thank the reviewer for insightful suggestions.*

Comment 1: Actually, there are lots of reports indicating that aqueous binder, such as, PAA, LA 132, modified β -Cyclodextrins, guar gum and so on, demonstrate better performances than traditional PVDF binder for elemental sulfur cathodes. If possible, please compare TG with aqueous binders.

Author reply: *Authors thank the reviewer for the suggestion on detailed comparison among the variously reported binders, especially organic and bio-based water-soluble binders, with that of TG. As per the reviewer suggestion, a detailed comparison was made among other reported organic binders, aqueous binders with the TG by considering four section headings, (i) organic binder, (ii) bio-based binder, (iii) cell parameter, and (iv) cell performance which were updated in the revised supplementary table 1 and 2.*

ACTION: Revised table 1 and 2 that compares TG binder with the reported organic and aqueous binders are included in the revised supplementary information and provided below.

In Supplementary Table 1, “Comparison of sulfur mass loading for variously reported binders in Li-S batteries (C-rate based on sulfur’s theoretical capacity 1.675 A g⁻¹).”

Binder (B)	Cell parameter				Cell performance						Ref.
	Electrode ratio (%) S/C/B	Sulfur load (mg cm ⁻²)	E/S ratio (μL mg ⁻¹)	Rate (C)	Areal Capacity (mAh cm ⁻²)		Cycle no.	Capacity retention (%)	Current collector		
					Initial	Final					
PVDF	7:2:1	2.0	-	0.4	1.8	1.62	50	90	Ni foam	6	
Gelatin	6.3:3:0.7	2.0	-	0.4	2.3	0.8	50	34	Al foil	7	
PAMAC	8:1:1	3.0	-	0.5	3.2	2.0	105	62.5	C-coated Al foil	8	
DCP	6.4:2.6:0.1	9.8	5	0.5	9.2	6.5	50	70.6	Ni foam	9	
CDp-N+	7:2:1	5.5	7	0.3	6.6	4.4	45	66.6	Al foil	10	
MPEII	5:4:1	6.5	-	0.05	7.52	5	70	66.4	Al foil	11	
PAMAM	8:1:1	4.0	40	0.05	4.56	2.8	100	61.4	-	12	
S/GG-XG	8:1:1	6.5	21	0.8	4.2	~4.7	90	101	-	13	
Carrageenan	-	5-10	-	0.01	~9	~5.8	100	64.4	-	14	
CMC-CA	8.5:4.5:5.5	10	4	1.0	~11.2	~5.7	50	50.8	Carbon paper	15	
PEI	6:3:1	8.6	16	0.05	9.7	6.4	50	65.9	Al foil	16	
Tragacanth gum	8:1.5:0.5	8.1 4.7	12- 15	0.1 0.1	10.8 9.7	10.02 6.3	100	92.7 64.9	Al foil Al foil	Our Work	

In Supplementary Table 2, “Comparison of Li-S battery performance and its stability using various bio-based and conventional binders (C-rate based on sulfur’s theoretical capacity 1.675 A g⁻¹).”

Type of polymer binder	Binder (B)	Cell parameter			Cell performance				Ref.	
		Electrode ratio (%) S/C/B	Sulfur load (mg cm ⁻²)	E/S ratio (μL mg ⁻¹)	Rate (C)	Specific capacity (mAh g ⁻¹)		Cycle no.		Capacity decay per cycle
						Initial	Final			
Organic	PVDF	7:2:1	2.4	50	0.5	788	322	100	0.49	17
	PAA	6:3:1	1.5	-	0.2	390	298	100	0.09	18
	PAA/ PEDOT:PSS	7:2:1	0.8	-	0.5	1121	830	80	0.36	19
	PVA	6:3:1	3.5	9	0.5	936	563	75	0.49	20
Bio-based	Gum arabic	8:2	1.5	100	0.2	1157	841	500	0.054	21
	Guar/ Xanthum	8:1:1	6.5	21	0.5	1000	724	150	0.18	10
	Na-alginate	5:4:1	1.5	-	0.5	776	508	50	0.53	22
	Gelatin	6.3:3:0.7	2.0	-	0.5	950	680	100	0.28	8
	Zein/CNF	6.38:2.41: 0.21	1.8	-	0.6	890	590	500	0.06	23
	SBR: CMC	5.4:3.6:1	1	-	0.2	867	612	150	0.55	24
	Chitosan	6.3:3:0.7	1-1.5	-	0.2	1145	780	100	0.36	25
	Gelatin/ Chitosan	6.3:3:0.7	1-1.5	-	0.5	857	675	200	0.09	27
	Soy protein- PAA	6.75:2.25: 0.1	1.5	10	0.3	790	595	200	0.12	26
	LA-132	7:2:1	2	80	0.5	901	470	100	0.43	27
	LA-133	5.4:3.6:1	1	-	0.2	1176	709	150	0.31	22
	Catechol- Chitosan sulfate	8.5:0.5:1	1.3- 1.5	15	0.2	785	605	400	0.045	28
	Laponite	8:1:1	1	-	0.5	810	595	500	0.05	29
	Polyurethane -Phytic acid	-	0.8- 1.5	25	0.5	1051	632	500	0.08	30
	Cyclodextrin -S composite	7:2:1	2.5	7	0.5	692	636	50	0.02	AFM
	Tragacanth gum	8:1.5:0.5	1.1- 1.4	15-17	0.2 1	1239 1236	976 865	200 300	0.13 0.12	Our work

Comment 2: High loading is a good result. What was the tap density of elemental sulfur cathode with TG binder?

Author reply: The authors sincerely thank the reviewer for suggesting an essential electrode parameter, tap density. Tap density was measured for the sulfur cathode employing all binders by considering the electrode powders ratio 8:1.5:0.5 of S/C, MWCNT, and PVDF/PEO/TG binder. For the measurement, the electrode constituents were gently intermixed for about 45 minutes to attain homogeneity and then transferred to a high precision 5 mL screw-type borosilicate transparent vial. Upon transferring powders, the borosilicate vial filled with electrode powders was mechanically tapped to a flat surface with a constant force of about a hundred repetitions. Upon completion of tapping, the amount of sample gathered at an exact minimum milliliter point was considered for calculating the tap density through the formula,

$$\rho^t = \frac{M}{V_t}$$

where, ρ^t is tapped density, M is powder mass, and V_t is the minimum volume occupied after tapping. Accordingly, the sulfur cathodes employing S/PVDF, S/PEO, and S/TG binder exhibited a tap density of 1.08 g cm^{-3} , 1.06 g cm^{-3} , and 1.08 g cm^{-3} , respectively.

Action: On page 12 in the revised manuscript, "The resultant method allowed the loading mass of sulfur to successfully exceed even above 12 mg cm^{-2} with a tap density of S/TG powders at 1.08 g cm^{-3} ."

In supplementary Table 7, "Tap density measurements for S/PVDF, S/PEO, and S/TG cathodes with electrode constituent ratio 8: 1.5: 0.5 (S/C: MWCNT: Binder)."

Test	S/C: MWCNT: PVDF		S/C: MWCNT: PEO		S/C: MWCNT: TG	
	Density (g cm^{-3})	Error	Density (g cm^{-3})	Error	Density (g cm^{-3})	Error
A	1.0862	0.00008	1.0643	0.00064	1.0863	0.0014
B	1.0853	0.00082	1.0623	0.00264	1.0848	0.0001
C	1.0883	0.00218	1.0668	0.00186	1.0822	0.0041
D	1.0861	0.00002	1.0647	0.00024	1.0857	0.0008
E	1.0847	0.00142	1.0666	0.00166	1.0855	0.0006
Average	1.08612	0.000904	1.06494	0.001408	1.0849	0.007

Comment 3: E/S ratio is an important parameter for elemental sulfur cathode. Does TG binder reduce E/S ratio remarkably?

Author reply: The authors thank the reviewer's excellent suggestion on one of the important parameters determining the electrochemical performance of Li-S batteries. Electrolyte to sulfur (E/S) ratio was determined for all the sulfur cathodes through two different experimental techniques,

- (i) Wettability test for electrodes
- (ii) Electrochemical studies of Li-S cells employing various sulfur cathodes with desired E/S ratio

For both the experiments, the electrolyte used was 1 M LiTFSI (lithium bis(trifluoromethane sulfonyl)imide) salt dissolved in an anhydrous mixture (1:1, v/v) of DOL (1,3 dioxolane) and DME (1,2-dimethoxymethane) with 2% lithium nitrate (LiNO₃).

Wettability test: First, the wettability tests were performed for all the sulfur cathodes by steeping them individually in an excess electrolytic solution. The respective mass change of the electrodes at 0 h and 24 h of steeping was measured to estimate the wettability of various sulfur cathodes. The S/TG cathodes showed a 211.94% increase in the overall mass of the electrode after 24 h, while the S/PVDF and S/PEO electrodes also showed an increase of 117.6% and 144.92 % electrode mass. The increased mass of S/TG electrodes exposes good wettability of macromolecular binder towards the electrolyte.

Electrodes \ Time	Mass at 0 h	Mass after 24 h	[%] increase
S/PVDF	5.1 mg	11.1 mg	117.6
S/PEO	6.9 mg	16.9 mg	144.92
S/TG	6.7 mg	20.9 mg	211.94

Electrochemical studies: Secondly, galvanostatic charge/discharge studies were performed for the Li-S cells employing S/PVDF, S/PEO, and S/TG electrodes. The mass of electrodes ranged 2.0 to 2.1 mg sulfur loading were assembled in a coin cell with a specified amount of electrolyte to sulfur (E/S) ratio of 6 μ L, 9 μ L, 12 μ L, and 15 μ L per milligram of active sulfur. Upon assembling, the cells were aged for about 2 h and were tested at a current rate of 1 C (1 C = 1.75 A g⁻¹) at room temperature. The S/TG electrodes exhibited specific capacities of 249 mAh g⁻¹ and 342 mAh g⁻¹ for the E/S ratio at 6 μ L and 9 μ L. Further, as the discharge/charge cycles continued, the capacity increased and attained a stable capacity of 478 mAh g⁻¹ and 500 mAh g⁻¹ at the end of the 100th cycle. Interestingly, the capacity of the S/TG electrodes at E/S ratio of 12 μ L to 15 μ L attained 593 mAh g⁻¹ and 920 mAh g⁻¹, which is higher than the capacity observed at 6 μ L and 9 μ L. However, as the cycles progressed, S/TG electrodes experienced a minor decay with a discharge capacity of 584 mAh g⁻¹ and 640 mAh g⁻¹ at the end of the 100th cycle. Considering the S/PVDF and S/PEO electrodes, there is a significant difference in the reversible capacity compared to the S/TG electrodes. At a higher E/S ratio of 12 and 15 μ L, the S/PEO electrodes showed better discharge capacities (than lower E/S ratio) at 280 mAh g⁻¹ and 412 mAh g⁻¹

with higher capacity decay. In the case of a lower E/S ratio of 6 μL and 9 μL , S/PEO electrodes exhibited initial discharge capacities of 62 mAh g^{-1} and 82 mAh g^{-1} , respectively, signifying that the PEO electrodes do not have sufficient electrolytes to allow the reactivity of Li^+ ion towards sulfur. For the S/PVDF electrodes, a reversible capacity of 341 mAh g^{-1} and 378 mAh g^{-1} was attained at E/S ratio of 12 μL and 15 μL ; upon lowering the E/S ratio to 6 μL and 9 μL , the electrodes experienced a low reversible capacity, but better than S/PEO electrodes. The results of E/S ratio along with wettability studies denotes that the S/TG electrodes are capable of providing access to Li^+ ions causing reactivity of sulfur active particles to a greater extent under minimal electrolytic condition.

Comparatively, the high wettability of S/TG electrodes experienced through the wettability test is corroborated with the results of electrochemical studies. The use of TG binder for sulfur cathode significantly promotes facile Li-ion transport due to lower Li-ion transfer resistance (Fig. 1e) and also facilitates direct contact between active material and electrolyte, which certainly improved the solvent percolation. As a result, the S/TG electrode promotes high accessibility for Li-ions to active sulfur particles even at limited electrolyte conditions.

Action: On page 14 in the revised manuscript, "Apart, the E/S studies for S/TG electrodes (2.7 mg cm^{-2}) at 6 and 9 $\mu\text{L mg}^{-1}$ (Fig. 4 g-i and Supplementary Fig. 10) exhibited specific capacities of 249 and 342 mAh g^{-1} , respectively for the 1st cycle. As the cycles progressed, the electrode underwent sustained wettability to deliver increased capacities of 478 (191%) and 500 mAh g^{-1} (146%) at the end of the 100th cycle. Under a similar E/S ratio, the S/PVDF and S/PEO electrodes delivered poor capacities of 280 and 82 mAh g^{-1} , respectively; however, increasing the E/S 12 $\mu\text{L mg}^{-1}$ allowed the electrodes to achieve better capacity but at the expense of 41.64% and 51.47% capacity decay. Conversely, at an E/S 12 $\mu\text{L mg}^{-1}$, the S/TG cathode features high capacity retention of 98.48% (Supplementary Fig. 10f-i). Moreover, lowering E/S causes an increase in polarization due to the nucleation and growth of Li_2S leading to sluggish kinetics, the S/TG electrodes too follow. Meanwhile, even at a low to marginal E/S ratio, the S/TG offers improved cyclability and capacity retention due to good electrode wettability (Supplementary Table 3)."

Figure 4g-i in the revised manuscript,

In Supplementary Figure 10,

Comment 4: How about the self-discharge property of elemental sulfur cathode with TG binder?

Author reply: We extremely thank the reviewer for an excellent comment on the self-discharge property of S/TG cathodes. The self-discharge property of all the sulfur cathodes was studied through galvanostatic charge/discharge cycling performed at two different current rates, low (0.2) and high (1) C rates. The self-discharge properties of Li-S cells employing various S/PVDF, S/PEO, and S/TG

electrodes were measured as a function of potential drop and time monitored throughout the resting time of 6 h – 12 h – 24 h (1 day)– 48 h (2 days)– 72 h (3 days)– 96 h (4 days) and the capacity retention. Since the high C rates undergo a fast discharge/charge process, the evolution of polysulfides would be much quicker; thus, the self-discharge tests were designed at low and high C-rates.

Low C-rate: The S/TG electrodes cycled at 0.2 C were able to maintain the output voltage of 2.37 V – 2.36 V – 2.36 V – 2.33 V until 6+12+24+48 h of resting time and cycling. Further, the S/PEO and S/PVDF electrodes too showed minor voltage losses from 2.37 V – 2.35 V – 2.36 V – 2.23 V and 2.37 V – 2.36 V – 2.34 V – 2.3 V, respectively under similar resting time. Further, all the electrodes witnessed voltage losses at 72 h and 96 h of resting, while S/TG electrodes exposed minimal loss than the counterparts. The self-discharge behavior is also notable from the discharge capacity observed after resting time, where S/TG electrodes witnessed a good Coulombic efficiency at 97.3 %, 97.1 %, 95.3 %, 91.8 %, 90.4 %, and 88.8 % after 6 h – 12 h – 24 h – 48 h – 72 h – 96 h of resting time. Comparatively, S/PVDF electrodes exhibited drastic decrease in Coulombic efficiency at 88.7 %, 86.8 %, 80.3 %, 65.4 %, 61.2 %, and 55.6%, while S/PEO electrodes are better than S/PVDF at 93.3 %, 91.4 %, 89.1 %, 83.9 %, 83.2 %, and 82.4 %.

High C-rate: The self-discharge property at high current rates showed an almost similar trend. Among the electrodes, S/TG electrodes are stable at high current rates. The output voltage observed for S/TG electrodes are 2.37 V – 2.36 V – 2.33 V, and 2.2 V after resting time of 6 h – 12 h – 24 h and 48 h, respectively. Further, the S/PEO electrodes showed minor voltage losses from 2.33 V – 2.34 V – 2.29 V, whereas S/PVDF electrodes showed 2.3 V – 2.24 V – 2.18 V at 6 h, 12 h, and 24 h of resting time. Further, at 48 h of resting, all the electrodes witnessed a voltage loss; however, the S/TG electrodes exposed minimal loss than the counterparts. The self-discharge behavior was also notable from the discharge capacity observed after resting time. The S/TG electrodes witnessed a good Coulombic efficiency at 91.9 %, 89.4 %, 85.3 %, and 83.3 % after 6 h – 12 h – 24 h and 48 h resting time. Comparatively, S/PVDF electrodes exhibited drastic decrease in Coulombic efficiency at 68.5 %, 65.07 %, 56.9 %, and 50.8 %, while S/PEO electrodes are better than S/PVDF at 91.3 %, 84.5 %, 56.9 %, and 64.6 %.

Action: Figure 4d-f in the revised manuscript,

On pages 13-14 in the revised manuscript, "The internal polysulfide shuttle upon resting Li-S cells were examined through self-discharge studies (Fig. 4d-f and Supplementary Fig. 8,9) at low (0.2 C) and high (1 C) current rates. The discharge profile depicted in Fig. 2d-f exposes minimal ΔV_x and limited discharge capacity losses after resting, and recovery cycle for S/TG than the S/PVDF and S/PEO electrodes, even open circuit potential (OCP) remained high for S/TG electrodes (Supplementary Fig. 8d). The Coulombic efficiency (CE) based on the respective discharge/charge of the n^{th} cycle (QD_n/QC_n) (Supplementary Fig. 8e,f) after resting remained steady for the S/TG electrode at 97.3% and 88.8% after 6 h and 96 h, respectively against 88.7% and 55.6% for S/PVDF and 93.3% and 82.4% for S/PEO electrodes. At a high current rate (Supplementary Fig. 9), the S/TG electrodes exhibited a reasonable CE of 83.3% after 48 h resting. Since the (ir-)recoverable capacity upon resting depends on the reaction of migrated soluble LiPS forming $\text{Li}_2\text{S}_2/\text{Li}_2\text{S}$ precipitate in the Li anode, the better recoverable capacity of S/TG electrodes is owed to good polysulfide anchoring of TG."

Supplementary Figure 8, (Low C-rate 0.2 C)

Supplementary Figure 9, (High C-rate 1 C)

Comment 5: It is difficult to understand TG working as flame retardant for elemental sulfur. Please further explain.

Author reply: The authors regret the difficulty caused in the explanation of the flame-retardant capability of the TG binder. Sulfur being a flammable solid and challenging to extinguish upon the fire, its use as active material in Li-S batteries requires high safety while at operation and in ideal condition. Thus, the flame-retardant ability being an essential criterion, sulfur cathodes fabricated using binder tragacanth and other organic PVDF and PEO binders were tested and compared. For the flame test, all

the sulfur cathodes were directly exposed to the flame and kept until the electrodes got ignited, where the time taken for the electrodes to burn/damage by flame is taken as a measure to validate the flame-retardant ability of various sulfur electrodes. Interestingly, the S/TG cathodes upon exposure to the flame got ignited and immediately self-extinguished. As the exposure to the flame continued from 1s to 4s and to 8s, there was no burning or fire from the S/TG electrodes. On the contrary, S/PVDF cathodes remained unextinguished until 4s and started to burn thereafter. More adverse occurred with S/PEO, which burnt vigorously. In order to understand the chemistry behind the flame-retardant characteristics, all the electrodes were characterized through X-ray diffraction in their fresh and after the flame test. The XRD results depicted in Supplementary Fig. 17 reveal that the TG binder is an amorphous polymer; upon ignition, it forms a thermostable polymer that is stable enough to act as a self-extinguishing layer to suppress the flame.

Action: The section corresponding to the flame-retardant ability of the TG binder was modified in the Methods section of the revised manuscript.

In the Methods section on page 26, "Flame-retardant property Pre-weighed calendared electrodes of $\phi 12$ mm were directly exposed to the flame emanant tip from the flame lighter. The flame-retardant property of the electrodes was estimated based on the time that the electrodes were exposed to the flame and remained until it started to burn, while the flame is supplied continuously during the process. The time recorded to self-extinguish to the burning of electrodes was validated for exposing the safety of the sulfur cathodes."

REVIEWERS' COMMENTS

Reviewer #2 (Remarks to the Author):

After carefully reading the revised manuscript, I was convinced that the manuscript now is ready for publication.

Reviewer #3 (Remarks to the Author):

Authors have made good responses and some suitable revisions according to the previous reviewers' suggestions. Suggest to accept after minor revision. It was interesting and strange that TG demonstrated good flame retardant performance for flammable elemental sulfur, even 5% percent content. Please add more mechanism explanation. 243 Wh/kg or 230 Wh/L is not a good result for Li/S battery, which currently already exceeds 400 Wh/kg for a punch cell. Suggest to not emphasize the energy density.

Response to Reviewers Comments

Reviewer #2: After carefully reading the revised manuscript, I was convinced that the manuscript now is ready for publication.

Author response: *The authors sincerely thank the reviewer and express their gratefulness about the reviewer's comment on being convinced with the revisions made to the manuscript.*

Reviewer #3: Authors have made good responses and some suitable revisions according to the previous reviewers' suggestions. Suggest to accept after minor revision.

Author response: *The authors express sincere thanks to the reviewer's comment about the good response and suitable revisions to the revised manuscript as per the previous review report. Also, the authors thank the reviewer for the present comments on the flame-retardant mechanism and energy density of the Li-S cell. As per the reviewer's suggestions, we have further elaborated the mechanism behind the flame-retardant ability of the TG binder and the limited the use of energy density values.*

Comment 1: It was interesting and strange that TG demonstrated good flame retardant performance for flammable elemental sulfur, even 5% percent content. Please add more mechanism explanation.

Author response: *The authors thank the reviewer's suggestion. Firstly, the composition of electrodes is considered important while considering the overall output in terms of energy and power density of a cell. Thus, the decreased content or lower ratio of passive components, i.e., materials that do not involve/contribute to the capacity- are highly required for increasing the energy density of the Li-S battery. Thus, the authors have undertaken the work in such a way as to reduce the ratio of passive components in the electrode while maximizing the sulfur active material to achieve the better activity. So, 5% of binder was considered for making the electrodes which were also tested for the flame-retardant ability.*

Secondly, various binders (PVDF, PEO, or TG) with 5% content in sulfur cathodes were tested for flame-retardant ability. The TG-based electrode demonstrated a good flame suppression than the other binders with similar binder content. At the same time, it is essential to mention that the TG-based sulfur cathodes suppress flame to a better extent, let us say 6-8 seconds of ignition (Fig. 7g), which is

considered as a step forward in controlling the damages due to fire of Li-S batteries, rather than being ignited immediately or supporting the flame leading to serious safety issues.

As per the reviewer's suggestion, the possible mechanism behind the flame-retardant ability of TG is deduced through X-ray diffraction, as previously provided in the original manuscript.

ACTION: The corresponding mechanism included in the revised supplementary information page 23 is as follows,

“Supplementary Note 2: Mechanism for flame-retardant nature

X-ray diffraction studies performed for the fresh and after-flame exposed sulfur electrodes are shown in Supplementary Figure 17. The XRD pattern for pristine binders shows crystalline nature for PVDF and PEO, while TG exposed an amorphous state. The electrodes fabricated with binders exhibited peaks corresponding to sulfur and carbon, while binder's peaks are noticeable only for the case of PVDF and PEO electrodes which is in agreement with the pristine binders XRD peaks. Moreover, the electrodes fabricated with TG do not show any crystal features to the TG binder. Upon flame-exposed, all the sulfur cathodes exposed peaks respective of aluminium current collector, however the PVDF and PEO-based electrodes lost their crystallinity upon ignition causing serious damage to the electrodes as they succumb to more-flammable characteristics. In the case of TG, the amorphous gum which possesses abundant -OH functional groups upon contact with the flame (temperature ~180.3 °C) tend to lose water molecules forming a stable protective layer which would have acted as a protection and thereby suppress the sulfur active material from reacting vigorously to the flame (Figure 7g and Supplementary Figure 15) which is why the TG-based electrodes showed good electrode architecture (Figure 7g). From the flame-retardant studies, it is known that the amorphous nature of tragacanth in conjunction with other electrode constituents remained passive to the flame, while other PVDF and PEO electrodes reacted to flame. Thus, the property of TG employed as a binder in sulfur cathode would have largely benefitted from the characteristics. In addition, the surface digital photographs of fresh and flame-exposed electrodes (Figure 7g) revealed an intact electrode characteristic for TG binder employed sulfur cathode among the PVDF and PEO sulfur cathodes. This intact feature is corroborated with an amorphous nature of TG (before and after flame) noticeable from the XRD, which is related to the formation of a passive layer upon interaction with flame.”

Comment 2: 243 Wh/kg or 230 Wh/L is not a good result for Li/S battery, which currently already exceeds 400 Wh/kg for a punch cell. Suggest to not emphasize the energy density.

Author response: *The authors agree with the reviewer's point that the commercial Li-S battery under precise cell engineering has surpassed 350-400 Wh/kg. As per the reviewer's suggestion, the values corresponding to energy/power density are limited in the manuscript. However, the authors would like to make it clear that the energy density values reported in this work are representative of the TG binder's feasibility under a flexible Li-S cell with standard laboratory conditions; which may not be compared with commercial-grade Li-S battery as it is well known that they are engineered at every aspect of cell component starting from active and passive material, electrode thickness, lithium coating, electrolyte with additive components with a specific purpose.*

ACTION: **The value of the volumetric energy density was removed in the revised manuscript.**